# Dietary Challenges for Parasitoid Wasps (Hymenoptera: Ichneumonoidea); Coping with Toxic Hosts, or Not?

**DOI:** 10.3390/toxins15070424

**Published:** 2023-06-29

**Authors:** Donald L. J. Quicke, Mostafa Ghafouri Moghaddam, Buntika A. Butcher

**Affiliations:** Integrative Ecology Laboratory, Department of Biology, Faculty of Science, Chulalongkorn University, Phayathai Road, Bangkok 10330, Thailand; d.quicke@email.com (D.L.J.Q.); mostafa.g@chula.ac.th (M.G.M.)

**Keywords:** Ichneumonidae, Braconidae, Papiliondae, Melitaeinae, secondary plant compounds, adaptation

## Abstract

Many insects defend themselves against predation by being distasteful or toxic. The chemicals involved may be sequestered from their diet or synthesized de novo in the insects’ body tissues. Parasitoid wasps are a diverse group of insects that play a critical role in regulating their host insect populations such as lepidopteran caterpillars. The successful parasitization of caterpillars by parasitoid wasps is contingent upon their aptitude for locating and selecting suitable hosts, thereby determining their efficacy in parasitism. However, some hosts can be toxic to parasitoid wasps, which can pose challenges to their survival and reproduction. Caterpillars employ a varied array of defensive mechanisms to safeguard themselves against natural predators, particularly parasitoid wasps. These defenses are deployed pre-emptively, concurrently, or subsequently during encounters with such natural enemies. Caterpillars utilize a range of strategies to evade detection or deter and evade attackers. These tactics encompass both measures to prevent being noticed and mechanisms aimed at repelling or eluding potential threats. Post-attack strategies aim to eliminate or incapacitate the eggs or larvae of parasitoids. In this review, we investigate the dietary challenges faced by parasitoid wasps when encountering toxic hosts. We first summarize the known mechanisms through which insect hosts can be toxic to parasitoids and which protect caterpillars from parasitization. We then discuss the dietary adaptations and physiological mechanisms that parasitoid wasps have evolved to overcome these challenges, such as changes in feeding behavior, detoxification enzymes, and immune responses. We present new analyses of all published parasitoid–host records for the Ichneumonoidea that attack Lepidoptera caterpillars and show that classically toxic host groups are indeed hosts to significantly fewer species of parasitoid than most other lepidopteran groups.

## 1. Introduction

As Bernays [1] astutely articulated, the act of feeding undertaken by lepidopteran larvae is fraught with peril. The range of challenges encountered by caterpillars of butterflies and moths (Lepidoptera) is extensive. These challenges encompass a multitude of top-down and bottom-up pressures, which exhibit temporal, spatial, and ontogenetic variations both between and within species [2]. Caterpillars have to overcome the diverse defensive mechanisms of potential host plants, which comprise an array of deterrents including (a) leaf trichomes, (b) surface waxes, (c) silica crystals, (d) allelochemical-producing glands or tissues, and (e) feeding-induced plant responses [3]. Caterpillars exhibit a repertoire of responses toward the defensive traits of their food plants. These responses entail the modification of plant tissue to evade chemical and physical plant defenses, relocation to feed on less-defended tissues, and the implementation of physiological or chemical mechanisms to process and tolerate defensive compounds [4,5]. In parallel, the evasion of top-down pressures holds significant importance for caterpillars, given their susceptibility to an extensive array of natural enemies, encompassing parasitoids, pathogens, and predators [6,7]. Although the significance of caterpillars’ responses to bottom-up attacks is widely acknowledged in the context of herbivore ecology and evolution, the present study focuses on reviewing the challenges of parasitoid wasps as top-down pressures with toxic caterpillar hosts.

Amidst the vast array of vertebrate and invertebrate species that prey upon caterpillars, it is widely recognized that insect parasitoids constitute a paramount source of mortality for the majority of phytophagous insect species [8]. Certain parasitoid wasps exhibit a parasitic relationship with caterpillars as an integral part of their reproductive cycle [8,9]. However, not all caterpillars are equally susceptible to parasitism by wasps and some species have evolved toxic defenses that make them unpalatable to predators and/or toxic to potential parasitoid wasp attackers [10]. The level of toxicity in caterpillars can vary greatly, and some species have even evolved the ability to switch on or off their toxic defenses depending on the presence or absence of parasitoid wasps. Unsurprisingly, parasitoid wasps that specialize on caterpillars have evolved a comparable range of behavioral, chemical, and morphological responses specifically tailored to counteract caterpillar defenses [11,12].

The relationship between parasitoid wasps and the toxicity of their caterpillar hosts is complex and has important implications for both the wasps and the caterpillars. On the one hand, the ability of caterpillars to defend themselves against parasitoid wasps can reduce the success rate of parasitism, which, in turn, can reduce the population size of the wasps. On the other hand, the population size of the wasps may also depend on the degree of polyphagy of the wasps. The presence of toxic defenses in caterpillars can also act as a selective pressure for the evolution of counter-adaptations in the parasitoid wasps. Parasitoid wasps have evolved a variety of mechanisms to survive within the body of toxic caterpillar hosts. Some species of parasitoid wasps are able to avoid, detoxify, or tolerate the chemicals produced by their hosts, while others are able to selectively target and avoid feeding on toxic hosts [13,14,15,16]. Additionally, some parasitoid wasps have developed physical adaptations such as thickened cuticles or specialized structures that prevent the toxic chemicals from reaching their vital organs. Furthermore, parasitoid wasps can also manipulate the physiology and behavior of their hosts, such as altering their feeding patterns or suppressing their immune system, to create a more favorable environment for their own survival [17,18]. The ability of parasitoid wasps to survive within toxic caterpillar hosts is a result of a complex interplay between biochemical, physiological, and behavioral adaptations. Overall, the relationship between parasitoid wasps and the toxicity of their caterpillar hosts highlights the importance of coevolutionary arms races [19].

It is well known that many insects have evolved means of being toxic, and this can afford them a great deal of protection from predators and has led to the evolution of many amazing examples of aposematism and of Batesian and Müllerian mimicry [20]. Much work has been conducted on the effects of prey insect toxicity on their predation by vertebrates, rather less on predation by invertebrates, and very little on the effects on insect relationships with parasitoids.

The synthesis of protective secondary plant compounds may be costly [21], and therefore, not surprisingly, they may have evolved to regulate the amounts they synthesize according to conditions, upregulating toxin production in response to herbivory [22,23]. Whilst this will aid in deterring feeding by non-specialists, it might benefit specialist herbivores that sequester the defensive plant chemicals for their own protection.

### 1.1. Latitudinal Trends

In the 1970s, the number of observations based principally on Malaise trap sampling at different latitudes revealed a rather unexpected trend [24,25,26,27]. Whereas nearly all groups of organisms have their peak species richness in the moist tropics, this did not seem to be the case for Ichneumonidae. Whilst no studies at the time considered its sister group, Braconidae, the general experience of entomologists did not notice anything odd about braconids. As there is no doubt that potential hosts of ichneumonids had their greatest diversity in the tropics, this begged explanations for why Ichneumonidae should display ‘anomalous diversity’. The incompleteness of parasitoid taxonomy and the various biases affecting whether or not a species is described mean that we likely cannot yet be absolutely certain that Ichneumonidae show actual anomalous diversity [28], but they do not seem to be as tropicocentric as braconids.

In the context of this review on toxins, the significance of ichneumonid anomalous diversity is underscored by several hypotheses put forth to elucidate its underlying causes. Notably, host toxicity emerges as a prominent and favored explanation among these hypotheses, thereby rendering it pertinent to the present discourse.

#### 1.1.1. Resource Fragmentation and Predation Hypotheses

Firstly, we will briefly discuss two other hypotheses that attempt to explain the apparent deficit of Ichneumonidae in tropical regions. The resource fragmentation hypothesis (RFH) was initially proposed by Janzen and Pond [26] and posits that as species richness increases, there is a non-linear relationship with the number of parasitoid species that can be sustained. This is because the richer an area is in potential hosts such as Lepidoptera species, the relatively rarer each will be, thereby impeding the evolution of specialist parasitoids. The predation hypothesis (PH) was put forward by Rathcke and Price [27] as a sort of adjunct to the RFH. It is based on the idea that overall predation pressure is higher in the tropics, with ants playing a major role in consuming exophytic Lepidoptera caterpillars. This predation pressure might further decrease the population densities of those parasitoids that survive to adulthood such that their populations are no longer viable due to Allee effects. They noted that parasitized caterpillars often behave differently from non-parasitized ones, and might be more sluggish and, therefore, possibly even more prone to predation.

#### 1.1.2. Nasty Host Hypothesis (NHH)

Gauld et al. [29] put forward a hypothesis proposing that the availability of potential hosts for parasitoids in the tropics is comparatively lower than in extra-tropical regions due to the greater chemical toxicity of the tissues of tropical hosts. This primary hypothesis is derived from the secondary hypothesis that tropical plants contain higher levels of plant secondary compounds, which may make the insects feeding on them more toxic to potential parasitoids. Although there is limited research on tri-trophic interactions in natural tropical ecosystems, the available evidence indicates that high levels of plant secondary chemicals in the diet of herbivores can have adverse effects on potential parasitoids. Gauld et al. [29] drew attention to the fact that the number of aposematic insects appears to be higher in the tropics. However, as there are more insect species overall in the tropics one would expect there to be larger numbers of toxic ones in absolute terms, and whether they are proportionately more common does not appear to have been formally tested.

Nevertheless, Gauld [29] presents intriguing instances that warrant consideration. For instance, *Enicospilus americanus*, a large ophionine ichneumonid moth parasitoid, demonstrates a propensity for parasitizing numerous species of saturniid moth caterpillars within extra-tropical regions of North America and southern South America [30]. Similarly, *E. texanus* exhibits a broad host range encompassing various *Hemileuca* species (Saturniidae) in the southern United States and northern Mexico [30]. However, it is noteworthy that neither of these "generalist" parasitoid species extends its distribution into the Neotropics, and except for some possessing toxins in their spines, the hosts do not sequester plant toxins. Notably, extensive rearing efforts involving over 1050 *Enicospilus* species in Costa Rica [data on BOLD] indicate a preference for a singular host or several remarkably similar hosts [31]. These observations align with the "nasty" host hypothesis, suggesting that *Enicospilus* species display a broader range of hosts when parasitizing caterpillars that feed on relatively non-toxic trees and shrubs found in extra-tropical forests. In contrast, when targeting caterpillars that feed on relatively toxic trees and shrubs in tropical forests, *Enicospilus* appears to be restricted to a single host species. Presumably, this restriction is due to specific micro-behavioral adaptations and/or metabolic abilities enabling *Enicospilus* to evade the toxins present in that particular host [31].

The NHH suggests that tropical resources for a parasitoid community are better defended than equivalent resources in extra-tropical ecosystems due to the accidental chemical processing or evolutionary development of defensive traits. Consequently, tropical parasitoids must either be more specialized if they feed on a single host species or more versatile if they feed on multiple species, as compared to their extra-tropical counterparts. This challenge may result in a decrease in the number of parasitoid species that can thrive in a given habitat.

#### 1.1.3. The NHH, RFH, and PH Evaluated

Sime and Brower [32] compared evidence for the various hypotheses that had been put forward to explain the apparent anomalous latitudinal diversity of Ichneumonidae, and concluded that the NHH provides a parsimonious explanation based on the limited evidence at hand for the diversity patterns of parasitoids associated with Papilionoidea. However, they maintained an agnostic stance regarding its overall superiority as a general causal theory. It is important to note that the authors did not intend to offer an explanation for the overall anomalous diversity of ichneumonids but rather focused on specific case studies where peculiar diversity patterns of host and parasitoid were identified, and relevant observations were made.

What the NHH does not explain, however, is why Ichneumonidae should be especially susceptible.

### 1.2. Safe Haven Hypothesis (SHH)

The impact of sequestered compounds on parasitoids remains poorly elucidated, yet two contrasting perspectives have been proposed. One perspective suggests that the effects on parasitoids will resemble those observed in predators, with sequestered compounds exhibiting toxicity towards parasitoids in a dose-dependent manner [33,34]. In contrast, an alternative view posits that the sequestration of plant compounds transforms the host into a refuge devoid of enemies for parasitoids. By sequestering compounds, hosts become protected from predators, and thus so is the parasitoid as well [35,36].

The "safe haven hypothesis", proposed by Lampert et al. [37], postulates that parasitoids capable of tolerating higher concentrations of sequestered compounds will benefit from the presence of such compounds in their host. A specific case illustrating SHH is observed in the North American catalpa sphinx hawkmoth, *Ceratomia catalpae* (Sphingidae: Sphinginae, Sphingini), whose caterpillars only feed on the Indian bean tree genus *Catalpa* (Bignoniaceae) and which sequester the iridoid glycoside catalpol from the food plant [15]. This host offers a protective refuge for the development of its microgastrine braconid parasitoid, *Cotesia congregata,* since the host caterpillars are strongly avoided by predators and hyperparasitoids specific to *C. congregata*.

Therefore, the utilization of plant allelochemicals by caterpillars as a means to render themselves unpalatable to predators might potentially be counterproductive if a parasitoid can colonize them as a host. In these cases, the sequestering herbivore serves as a "safe haven" for the development of its parasitoid [38,39]. This is effectively the opposite of the predation hypothesis, and which is more important might perhaps depend on the most important predators, i.e., birds or ants. The SHH has been corroborated by investigations demonstrating the improved performance of some parasitoids when reared from more toxic individuals of a particular host species [16,40].

According to Barbosa’s hypothesis [13], specialized parasitoid wasps such as braconids and ichneumonids possess a higher tolerance towards the chemical compounds associated with their hosts, while generalist flies like tachinids cannot. Mallampalli et al. [41] conducted a study demonstrating that the tachinid fly, *Compsilura concinnata*, exhibited no negative effects when its host caterpillars were exposed to condensed tannins or the iridoid glycoside catalpol. This observation led them to propose that generalist flies may possess inherent resistance mechanisms against allelochemicals, thereby challenging the notion that these compounds universally impact their fitness.

## 2. The Toxins

Most of the toxins we consider here are secondary plant compounds, although some toxic hosts are capable of toxin synthesis de novo. Collectively, plants devote a lot of resources to the production of these compounds, which play important roles such as defense against pathogens and herbivores and protecting leaves against the damaging effects of UV radiation, as well as other things. Indeed, up to 30% of the extractable dry mass of terrestrial plants may be composed of secondary plant compounds [42].

In order for herbivorous insects to feed safely on plants that contain toxins, they must have one or more specific adaptations: (1) complete non-absorption in the first place [43]; (2) mutations to the target site(s) of the toxin that render those sites less susceptible [44]; (3) rapid detoxification through enzymic or blocking action; (4) carrier molecules that enable the safe storage of the compounds; (5) excluding the toxins from sensitive tissues and, in some cases, (6) carrier molecules that prevent the enzymatic activation of the toxin in the herbivore itself (see Section 2.6 below) [45]; or (7) having a gut environment that prevents the formation of the toxin from precursors [46]. For example, in the case of cyanogenic glycosides, which are enzymatically broken down to produce HCN, the latter can be inactivated by the insect by enzymatically combining it with cysteine [47].

The effects of toxic plant secondary compounds on herbivores and their parasitoids are often somewhat more complex than simply killing the parasitoid. Several studies have compared the effects of specific classes of toxins between specialist and generalist herbivores and their parasitoids [36,48,49].

### 2.1. Aristolochic Acids (AAs)

Caterpillars of most troidine and some parnassiine papilionid butterflies feed exclusively on members of Aristolochiaceae and sequester AAs and related compounds [50]. Many of the former, in particular, are highly toxic and are therefore avoided by predators. Very few parasitoids are able to tolerate these chemicals.

The well-known American species, *Battus philenor,* is the model in a mimicry complex [51,52]. Direct evidence supporting the role of AAs as defensive compounds, specifically in deterring predators when topically applied to palatable organisms, remains limited. Sime [53] demonstrated that AAs were effective in deterring the attack by the parasitoid *Trogus pennator* (Ichneumonidae), a generalist parasitoid, on larvae of *B. philenor* (see Section 4.2.1 below). Additionally, AAs have been detected in various parts of unpalatable Troidini and Zerynthiini, including the osmeterium, integument, hemolymph, and whole body [54,55]. The high concentration of AAs in the integument and osmeterium suggests their defensive role [55]. Another example is the gregarious koinobiont endoparasitoid *Glyptapanteles aristolochiae* (Braconidae: Microgastrinae), which parasitizes *Pachliopta hector* (Papilionidae: Papilioninae) larvae feeding on *Aristolochia indica*. However, it appears that this parasitoid has developed mechanisms to overcome the sequestered AAs [56].

### 2.2. Cardiac Glycosides (Cardenolides)

Cardenolides are a class of heart poisons produced by milkweed plants (Apocynaceae: Asclepiadoideae) and are famously sequestered by caterpillars of the toxic milkweed (monarch) butterfly, *Danaus plexippus*, in N. America, as well as other members of the subfamily Danainae worldwide. Cardenolide itself is a steroid, but is normally present in plants as a sugar derivative, i.e., glycoside. Cardenolides are highly specific inhibitors of the electrogenic sodium–potassium exchange pump enzyme, Na+/K+-ATPase, whose activity creates the resting potentials across all cell membranes but is particularly important in the case of nerve cells. Therefore, inhibition of this enzyme severely disrupts nerve and muscle activity [57].

Cardiac glycosides are known to be produced by members of 12 plant families, but they dominate in Apocynaceae, being present in 31 different genera [57]. It should be noted that whilst this is a minority of genera in the family, most others produce alkaloids, so the majority of Apocynaceae are toxic.

### 2.3. Cyanogenic Glycosides (CGs)

Cyanogenic glycosides are known in more than 2500 species of plants distributed among approximately 100 plant families, notably Fabaceae, Lauraceae, Cecropiaceae, Meliaceae, Passifloraceae, and seed kernels of Rosaceae [58,59]. They are typically synthesized from the amino acids tyrosine, phenylalanine, valine, leucine, and isoleucine [60]. The synthetic pathway involves P450 enzymes and a glucosyl transferase [47], and Jensen et al. [61] showed that the entire biosynthetic pathway of the CGs in the burnet moth *Zygaena filipendulae* is encoded by just three genes (YP405A2, CYP332A3, and UGT33A1).

The most common, indeed almost universal, CGs are linamarin and lotaustralin [62,63,64]. The release of poisonous hydrocyanic acid from these depends on the presence of highly specialized glucosidases and is also pH-dependent.

While quite a few insects that feed on cyanogenic plants sequester the compounds, some have evolved pathways to synthesize them themselves. It is possibly that evolution first led to the ability to chemically manipulate them and, later, mutations to the ability to make their own. Witthohn and Naumann [65] report detecting 𝛽-cyano-L-alanine in Papilionidae, Pieridae, Lycaenidae, Hesperiidae, Lymantriidae, Arctiidae, Notodontidae, Megalopygidae, Limacodidae, Cymatophoridae, Noctuidae, Geometridae, and Yponomeutidae, as well as the well-known examples Nymphalidae, Zygaenidae, and Heterogynidae, and this seems to be strong evidence for the capability of all of these to be cyanogenic.

CGs are not only the main chemical defense mechanism in Zygaenidae, but various Heliconiine butterflies synthesize their own. Nahrstedt and Davis [62] showed that even more basal members of the Heliconiinae, i.e., *Agraulis*, *Dryas,* and *Cethosia*, as well as the famous *Heliconius* itself, were all able to synthesize linamarin and lotaustralin, although to a lesser extent.

### 2.4. Fouranocoumarins

Fouranocoumarins are a group of secondary plant compounds formed from the fusion of a furan ring with a coumarin. The three rings may be in a straight line, and these are called linear fouranocoumarins (e.g., psoralen), or the furan ring may be joined at an angle to the long axis of the coumarin, and these are called angled fouranocoumarins (e.g., angelicin) [66]. They are produced mainly by members of Apiaceae and Rutaceae, although they are also present in the milky sap of fig trees (*Ficus* spp., Moraceae).

Many fouranocoumarins, especially linear ones, are noted for their toxicity to mammals and, interestingly, they are photo-activated. Upon ingestion or contact, the compounds enter epithelial cells, and then if in the nucleus and exposed to sunlight (the UV portion thereof), they bond to the cell’s DNA, leading to cell death. This, in turn, causes inflammation via activation of the arachidonic acid cascade, and in humans, this results in the unpleasant and often serious condition called phytophotodermatitis [67]. However, a few herbivorous insects specialize in plants containing these compounds [68].

Many Papilionidae feed on Apiaceae that contain fouranocoumarins but do not sequester them, and thus possess mechanisms to metabolize them. In the N. American black swallowtail, *Papilio polyxenes*, which feeds upon fouranocoumarin-containing Apiaceae and Rutaceae, detoxification is achieved by two cytochrome P450 monooxygenases (CYP6B1 and CYP6B3) in the caterpillar’s midgut [69,70,71]. Petersen et al. [72] have more recently shown that the fat body also metabolizes significant amounts of the linear furanocoumarins bergapten and xanthotoxin after larvae feed on xanthotoxin.

Plants of the genus *Apium* (Apiaceae) produce the linear furanocoumarins psoralen, bergapten, and xanthotoxin. Reitz and Trumble [73] investigated the effects of these in the pest cabbage looper caterpillar *Trichoplusia ni* (Noctuidae: Plusiinae, Argyrogrammatini) and on its polyembryonic encyrtid parasitoid *Copidosonma floridanum* (Chalcidoidea). Host and parasitoid larval mortality were found to be positively correlated with linear furanocoumarin concentration but the effect on the parasitoid was far greater.

### 2.5. Glucosinolate–Myrosinase System

Glucosinolates comprise a sugar moiety with a side chain containing both nitrogen and sulfur atoms (thioglycosides). They are produced in various amounts by most members of the plant order Brassicales, which includes many common edible plants (cabbage, broccoli, horseradish, mustard, capers, and black pepper) [74]. They contribute to the flavor and, in low concentrations, stimulate appetite. They have long been recognized as being important in defending various members of the Brassicaceae from herbivory [75]. The defensive efficacy of glucosinolates primarily stems from the isothiocyanate products generated through myrosinase (β-thioglucosidase)-catalyzed hydrolysis upon tissue damage [76]. Isothiocyanates, known for their toxicity to various organisms including herbivorous lepidopterans, are well documented [75].

Two mechanisms of Lepidoptera caterpillars dealing with glucosinolates have been described. The well-known cabbage pest diamondback moth, *Plutella xylostella* (Plutellidae: Plutellinae) possesses glucosinolate sulphatase, which desulphates glucosinolates that can no longer be modified by myrosinase to produce toxic products [77,78]. White butterflies (Pieridae: Pierini) possess the so-called nitrile specifier protein, which diverts glucosinolate hydrolysis toward the formation of nitriles rather than reactive isothiocyanates [79]. Thus, *P. xylostella* caterpillars are hardly chemically protected [78], facilitating their attack by numerous parasitoid species, the notable ones in terms of pest management being various species of the campoplegine ichneumonid genus *Diadegma* [80] and a few species of microgastrine braconids [81] such as *Cotesia vestalis*.

### 2.6. Iridoid Glycosides (IGs)

IGs are derived from terpenoids (cyclopentanoid monoterpenes) and named after the defensive compounds iridoidial and iridomyrmecin, which, in turn, are named after the ant genus *Iridomyrmex* [82]. Irido- means rainbow. They can be subdivided into four separate classes: iridoid glycosides, non-glycosidic (aglycone) iridoids, secoirioids, and bisiridoids [45]. IGs are synthesized by various plants via the malvonic acid pathway [83]. An inventory of iridoid glycosides (IGs) has revealed the identification of over 600 distinct compounds derived from 57 plant families [82]. Table 1 lists some of the named IGs and their sources. Notably, a considerable proportion of these compounds exhibit a bitter taste, and certain IGs have been documented as toxic to livestock [84]. IGs become activated only after enzymatic cleavage by β-glucosidases in herbivorous insects’ gut. The enzyme can be endogenous to the herbivore or may be co-ingested from the food plant (or both). The aglycone cleaved off by the β-glucosidase non-specifically crosslinks proteins and thus inhibits many enzymes. The effects are often to reduce the protein digestion and nitrogen uptake efficiency of the caterpillar [85]. Since β-glucosidase is also required, effective plant toxicity is dependent on concentrations of both this enzyme and its substrate.

Iridomyrmecin is actually a nonglycosidic iridoid since it lacks a sugar moiety. The IG content of various co-occurring *Euphydryas* (Nymphalidae: Nymphalinae, Melitaeini) may be responsible for their generally similar appearance, especially wing undersides—a Müllerian mimicry system, although rather than convergent coloration, it is likely that natural selection acted against deviation from an ancestral pattern [20]. Iridoid glycosides (IGs) are commonly recognized as feeding deterrents or toxic substances for non-specialist herbivorous insects. Nayar and Fraenkel [86] suggested that they might be responsible for the host specificity of the North American hawkmoth, *Ceratomia catalpae* (Sphingidae: Sphinginae, Sphingini), whose caterpillars only feed on the Indian bean tree genus *Catalpa* (Bignoniaceae).

The geometrid moth, *Meris paradoxa*, investigated by Boros et al. [87] is interesting because the highly aposematic caterpillars sequester the IG antirrinoside in large quantities, up to 11% of their dry weight, from their natural food plant, *Maurandya antirrhiniflora* (Scrophulariaceae). However, the adults are cryptically colored and contain only trace quantities of antirrinoside. Some of the IG is lost in the last larval molt and some in the meconium, but the data suggest that most of it is metabolized before the moth ecloses. The same is true of *Ceratomia catalpae*. *M. paradoxa* does not appear to be the host of any parasitic Hymenoptera.

IGs generally have a bitter taste to humans and are potent antifeeding agents against most insects [88] and also birds [45]; thus, many herbivorous insects that can feed upon and sequester them are aposematic.

### 2.7. Pyridine Alkaloids (Nicotine)

Nicotine is a di-nitrogen alkaloid based on a pyridine ring that is produced, along with a few related compounds, by tobacco, *Nicotiana tabacum* (Solanaceae), and is of huge commercial importance. Nicotine acts principally as an agonist at most nicotinic acetylcholine receptors (nAChRs) in the vertebrate peripheral and central nervous systems (CNS), including the excitatory nAChRs at neuromuscular junctions. The nAChRs of insects are almost entirely located in the CNS since their excitatory neuromuscular transmission is mediated by L-glutamate. Research in this area has led to the development of the neonicotinoid class of insecticides [89].

### 2.8. Pyrrolizidine Alkaloids (PAs)

PAs are heterocyclic secondary metabolites characterized by a pyrrolizidine motif [90] and are synthesized by plants as defensive compounds to deter herbivory. More than 660 distinct PAs have been chemically identified from an estimated 6000 different plant species [91]. Common ragwort, *Jacobaea* (=*Senecio*) *vulgaris* (Asteraceae), is highly toxic to livestock and in some countries, there are laws requiring landowners to remove it. However, the common practice of simply pulling off the above-ground part can be counterproductive as the base of the stem will simply send up more shoots, usually four instead of one (M. Crawley, pers. comm.). PAs may be passed by the female insect into her eggs, thereby providing them (and possibly first-instar larvae) with protection against early parasitoids such as trichogrammatid (Hymenoptera, Chalcidoidea) [92].

Most PAs, especially their *N*-oxide forms, appear to be able to bind with mammalian (pig) muscarinic acetylcholine receptors but not with nicotinic ones, and also bind to serotonin receptors, but less effectively to adrenergic receptors [93]. In insects, acetylcholine is one of the most important central nervous system neurotransmitters and muscarinic and mixed muscarinic–nicotinic types predominate. Some PAs and their *N*-oxides (PANO) can cause liver toxicity, as well as their metabolites causing DNA damage. Therefore, the occurrence of PA/PANO in foodstuffs has recently been regulated in the European market for certain foodstuffs, with an acceptable upper limit of 750 μg/kg for fresh or frozen borage leaves and 1 mg/kg for dried borage, *Borago officinalis* L. (Boraginaceae). Since the toxic effects of PAs in mammals often take days or weeks to become apparent, Hartmann and Witte [94] argued that their main role is likely as antifeedants to insectivorous herbivores.

Interestingly, PA consumption is of great importance to various species of Danaini and Ithomini nymphalid butterflies and tiger moths (Erebidiae: Arctiinae) as precursors of male sex pheromones [95,96].

### 2.9. Tannins

Tannins are ubiquitous secondary plant compounds that reduce protein metabolism in animals feeding upon them, slowing their development and likely reducing their overall food consumption and survival [97]. In terrestrial plants, tannins are the fourth most important group of compounds after cellulose, hemicelluloses, and lignin [42]. Although it has long been assumed that tannins, through forming complexes with other proteins, reduce enzymatic activity, the situation is likely far more complex [42].

Tannins may have both direct and indirect effects on insect growth. As a result of feeding on tannin-rich food, the tissues of herbivorous insects will also come to contain tannins. Yang et al. [97] demonstrated that tannic acid in the diet of host larvae has detrimental effects on the fitness of the microgastrine braconid wasp *Microplitis mediator*, and Roth et al. [98] found similar effects with the gypsy moth, *Lymantria dispar*, and its microgastrine parasitoid *Cotesia melanoscelus*. This parasitoid species is a koinobiont endoparasitoid attacking various pest Lepidopterans, including *Helicoverpa armigera* (Lepidoptera: Noctuidae). The effect of tannins on the parasitoid’s fitness was primarily a reduction in larval survival, and the body mass of those emerging parasitoid adults was reduced. Interestingly, adult wasps also exhibit direct consumption of tannic acid in their food, further influencing the development and fitness of the parasitoid.

### 2.10. Tropane Alkaloids

Named because the toxin molecules contain a tropane ring, these secondary plant compounds are mainly produced by various members of Solanaceae and also by Erythroxylaceae (which includes *Erythroxylum coca*, the cocaine-producing coca plant). They include some of the better-known poisons and drugs such as atropine from deadly nightshade (*Atropa belladonna*), hyoscyamine from henbane (*Hyoscyamus niger*), mandrake (*Mandragora officinarum*), and the sorcerers’ tree (*Latua pubiflora*), and scopolamine from henbane and *Datura* species. As with PAs, tropane alkaloids display a strong binding to muscarinic acetylcholine receptors [99].

**Table 1 toxins-15-00424-t001:** Major classes of sequestered secondary plant compounds and host de novo-synthesized toxins whose effects on parasitoids has been investigated.

Toxin Group	Named Examples	Plant Group/Families Containing Them	Lepidoptera Groups Feeding on Them	Reference(s)
Aristolochic acid and related compounds		Aristolochiaceae	Papilionidae: Parnassinae and Troidini	[50,56,100]
Cardiac glycosides (Cardenolides)	Digitoxin Digoxin Neoconvalloside Ouabain	Apocynaceae Asparagaceae Crassulaceae Moraceae Solanaceae	Erebidae: Arctiinae: Arctiini and Ctenuchini Nymphalidae: Danainae: Danaini	[101,102]
Cyanogenic glycosides (CNglcs)	Linamarin Lotaustralin Gynocardin	de novo (endogenous) biosynthesis Fabaceae Achariaceae Passifloraceae	Nymphalidae: Acraeinae and Heliconiinae Zygaenidae	[103,104,105]
Furanocoumarins	Psoralen Bergapten Xanthotoxin Angelicin	Apiaceae Moraceae (*Ficus*) Rutaceae Also, some widely distributed Fabaceae and Moraceae (latex)	Papilionidae	[66,68,73,106]
Glucosinolates	Sinigrin	Brassicaceae and other Brassicales, e.g., Capparaceae	Pieridae: Pierinae	[75,107,108,109]
Grayanoid glycosides		Ericaceae	Geometridae: Ennominae	[110,111]
Iridoid glycosides (IGs)	Aucubin Antirrhinoside	Scrophulariaceae (e.g., *Maurandya*, *Rhinanthus, Buddleja*) Cornaceae Orobanchaceae (*Melampyrum*) Rubiaceae	Geometridae (*Meris* sp.); Erebidae: Lymantriinae (*Ivela auripes*)	[87]
	Catalpol Catalposide	Bignoniaceae (Tecomeae, e.g., *Catalpa*) Orobanchaceae	Sphingidae Nymphalidae: Melitaeini	[15,35,112,113]
	Agnuside	Lamiaceae (*Vitex agnus-castus*)		[82]
	Amarogentin Gentiopicroside	Gentianaceae *(Gentiana*)	Nymphalidae: Melitaeini	[114]
	Asperuloside	Daphniphyllaceae (*Daphniphyllum*) Plantaginaceae Rubiaceae		[48]
	Loganin	Loganiaceae (*Strychnos nux-vomica* and spp.)	Lycaenidae	
	Macfadienoside	Orobanchaceae (e.g., *Castilleja*)	Nymphalidae: Melitaeini	[115]
	Plumeride Plumericin	Apocynaceae (*Plumeria*, *Himantanthus*)	Noctuidae (*Spodoptera frugiperda*) Sphingidae (*Pseudosphinx tetrio*)	[116]
	Digitoxin Digoxin	Plantaginaceae (*Digitalis*)	Nymphalidae: Heliconiinae: Melitaeini	[117,118,119]
Phenanthroindolizidine alkaloids		Moraceae (*Ficus*)	Erebidae: Aganinae	[120]
Phenolic compounds		Lichens	Erebidae: Arctiinae: Lithosini	[121]
Pyridine alkaloids	Nicotine Nornicotine Lobeline Anabasine	Solanaceae (*Nictotiana*) Campanulaceae (*Lobelia*)	Sphingidae (*Manduca sexta*)	[19,122]
Pyrrolizidine alkaloids (PAs)	Calotropin	Apocynaceae (especially Asclepiadaceae)	Nymphalidae: Danainae: Danaini	[123]
	Lycopsamine Senecionine	Asteraceae	Nymphalidae: Danainae: Ithomini Erebidae: Arctiinae	[96,124]
		Boraginaceae		[125]
		Fabaceae (Crotalarieae)	Erebidae: Arctiinae	[126]
		Orchidaceae		
		Solanaceae		[127,128]
		Some Convolvulaceae	Nymphalidae: Heliconiinae: Acraeini	[129]
	Thesinine	A few Poaceae		[130]
Pseudocyanogens	Cycasin	Cycadaceae	Lycaenidae (*Eumaeus*) Nymphalidae: Morphinae: Amathusiini: *Taenaris*	[131]
Quinolizidine alkaloids	Sparteine Cytisine	Fabaceae (*Genista*)	Pyralidae (*Uresiphita reversalis*)	[132]
Steroidal glycoalkaloids	α-Tomatine	Solanaceae (*Solanum*)	Noctuidae (*Heliothis zea*)	[133]
Tannins		Universal		[97]
Tropane alkaloids	Atropine Hyoscyomine Scopolamine	Solanaceae	Nymphalidae: Danainae: Ithomini Erebidae (Lymantriinae) Sphingidae	[134]

## 3. The Host

Although there are exceptions, the majority of highly unpalatable, chemically protected Lepidoptera are aposematic as adults, and often also as larvae [132] and sometimes as pupae [110]. They have been the source of much evolutionary research and insight because of the spectacular Batesian and Müllerian mimicry systems that the adults are often involved in [20]. Warning coloration is, of course, hardly effective at night and therefore most aposematic insects are diurnal, and in the case of Lepidoptera, not just butterflies but also various groups of day-flying moths. Interestingly, although we highlight some particular groups of butterflies as being particularly protected, Marsh and Rothschild [135] revealed several interesting insights into the toxicity patterns of aposematic Lepidoptera including the diverse nature of toxicity across different species, with variations observed in relation to larval diet, sexual dimorphism, color variations, and the retention of toxicity under different conditions.

Several groups of toxic and usually warningly colored (aposematic) moths used to be placed in their own separate families, e.g., tiger moths and allies in Arctiidae, Ctenuchidae, Syntomidae, and Aganaidae. Molecular phylogenetics has now shown that they are all members of Erebidae, which itself was previously mixed with the very large family Noctuidae, although both still belong to Noctuoidea. Syntomines and ctenuchines are now in Arctiinae, whereas the aganaines are now regarded as a separate, but still fairly closely related, subfamily of Arctiinae [136]. 

### 3.1. Erebidae

#### 3.1.1. Erebidae (Aganainae)

Aganainae are a small subfamily of moths in the family Erebidae, likely close to but separate from Arctiinae. In the earlier literature, they were usually regarded as the separate family Aganidae. The adults of this subfamily are typically large and aposematic (Figure 1A–D), like the related tiger moths, and this also applies to their large caterpillars. Many of them feed on poisonous host plants and acquire toxic cardenolides that make them unpleasant to predators [137]. 

The subfamily includes nine genera, all restricted to the Old World tropics, with *Asota* (Figure 1A,B) being the largest with more than 50 species, and commonly encountered at light traps. Many species of *Asota* are specialists on just a few species of *Ficus* (Moraceae).

Volf et al. [138] studied the defenses and caterpillars associated with 21 sympatric *Ficus* species in New Guinea and found that most generalist herbivores concentrated on hosts with low protease and oxidative activity, but the highly specialized *Asota* moths used alkaloid rich plants. Fontanilla et al. [139] performed a study on the chemicals of *Asota* species, their host plants, frass, etc., using UHPLC-MS/MS, and found a total of 43 different alkaloids. Some leaf alkaloids were excreted in frass or found in caterpillars and adult moths. Those alkaloids that were found in insect tissue were shared across moth species even though their caterpillars fed on different *Ficus* species, indicating that a specific subset of plant alkaloids have a direct ecological function, and that these roles were conserved across *Asota* species. They also found two novel indole alkaloids in both *Asota* caterpillars and adults, but not in leaves or in caterpillar frass, and concluded that there were likely synthesized de novo by the moths or by their microbiota.

#### 3.1.2. Erebidae (Arctiinae, Including Former Ctenuchidae and Syntomidae)

The tiger and footman moths used to be classified in the separate family Arctiidae, but they are now treated as a subfamily within Erebidae. Worldwide, the subfamily comprises approximately 11,000 species [140]. The former Synomidae are now considered a tribe within Arctiinae and Ctenuchidae a subtribe within Arctiini [140,141]. Many of the larger species display aposematic coloration (Figure 1E–G and Figure 2). 

The subfamily is currently classified into three tribes: Arctiini, Lithosiini, and Syntomini. Arctiini are further subdivided into nine subtribes, viz., Arctiina, Callimorphina, Ctenuchina, Euchromiina, Micrarctiina, Nyctemerina, Pericopina, Phaegopterina, and Spilosomina. However, the current state of Arctiinae phylogeny is still somewhat unsettled and the validities of some of these may change in the future [140].

Arctiini include most of the larger-bodied tiger moths, such as the well-studied garden tiger moth, *Arctia caja* (Arctiina) (Figure 1E), speckled footmen moth, *Utethesia* spp. (Callimorphina) (Figure 1G), and cinnabar moth, *Tyria jacobaeae* (Callimorphina) (Figure 2).

Caterpillars of the European garden tiger moth, *Arctia caja,* can sequester large quantities of various CGs, Pas, and other toxic compounds from their food plants, as well as synthesizing several defensive compounds themselves (e.g., acetylcholine, histamine, and a toxic protein). If forced to consume cannabis plant leaves, *Cannabis sativa,* not a natural host plant, they can nevertheless sequester ∆^1^-tetrahydrocannabinol from them [142]. These plant-derived toxins are mostly deposited in their cuticle [143].

Unlike those of most other groups of toxic Lepidoptera, caterpillars of some larger arctiines are highly polyphagous and often wander around on the ground from one small plant to another. They can feed on plants protected by both cardiac glycosides (CGs) and pyrrolizidine alkaloids (PAs) as well as other non-defended species such as members of Rosaceae [144]. Therefore, the level of toxicity of the caterpillars is likely quite variable both between individuals and at different times for the same individual as it develops and moves between food plants. Zaspel et al. [140] showed that Arctiinae likely evolved to be specialists on PA-containing foodplants early in their evolution and that facultative feeding on pharmacologically active plants likely evolved more recently. 

There appear to be multiple trade-offs when it comes to parasitism and PAs. Some research has found that dietary PAs in *Plagiobothrys arizonicus* (Boraginaceae) may have a negative effect on the immune system of the common North American tiger moth *Apantesis* (=*Grammia*) *incorrupta* and, consequently, would be less acceptable for feeding early on in the infection, when the encapsulation of the parasitoid is most crucial [145]. In this moth species, improved resistance to parasitism by the tachinid fly *Exorista mella* was a result of increased dietary intake of PAs [146], which, in turn, resulted from a change in caterpillar gustation in response to parasitism by this fly [147]. Caterpillars of some species are known to change their dietary preference in response to parasitization [146]. Indeed, Singer et al. [146] showed that the change differs depending on whether the parasitoid is a hymenopteran or a dipteran (Tachinidae). The important question that does not seem to have been answered is whether the dietary change is actually capable of killing the parasitoid larva and thus allowing the host to complete development and ultimately reproduce.

Arctiine larvae are variable in appearance. Some species are highly aposematic, e.g., those of the Eurasian cinnabar moth, *Tyria jacobaeae* (Figure 2A), a specialist feeder on ragwort *Jacobaea vulgaris*, which produces the poisonous alkaloid senecionine. In contrast, the larvae of many others are densely hairy, sometimes also with some bright spots, e.g., the so-called woolly bear caterpillars of carious *Arctia* species. In these latter cases, the hairs play an important role in defense against predators as they can cause skin irritation. The hairs potentially also serve as protection against attack by some non-specialist parasitoids.

The crimson-speckled footman, *Utetheisa pulchella*, (Figure 1G) is a major herbivore on heliotrope (*Heliotropium* spp., Heliotropiaceae) in the Mediterranean region, a generally toxic plant that is rich in abundant pyrrolizidine alkaloids [148]. Both records of ichneumonoid parasitoids of *Utetheisa pulchella* in the comprehensive Taxapad database of the superfamily up until 2015 [149] and the single one for the *U. ornatrix*, the N. American bella moth, *U.* whose larvae feed on *Crotalaria* (Fabaceae), involve microgastrine braconids.

Larvae of the well-known, aposematic N. American polka-dot moth *Syntomeida epilais* now feed widely on introduced oleander plants, *Nerine* (Apocynaceae) but its native food plant is likely the apocynacean vine *Echites umbellatus*. Interestingly, Rothschild et al. [150] showed that they sequester the cardiac glycoside oleandrin, which is abundant in *Nerine* but not normally sequestered by other insects and appears absent from *Echites*. Taxapad [149] includes no ichneumonoid host–parasitoid records for *Syntomeida*, which, given its abundance, suggests that it might almost be in enemy-free space, although McAuslane and Bennett [151] did find parasitism of its caterpillars by *Brachymeria incerta* (Chalcidoidea: Chalcididae) and the tachinid flies *Chetogena* (=*Euphorocera*) *floridensis*, *Lespesia aletia*, and an unidentified *Lespesia* species.

Spilosomina is a large subtribe with approximately 110 genera worldwide (Figure 1F) that are commonly called woolly bears or white tigers. It includes some pest species such as the highly polyphagous fall webworm moth, *Hyphantria cunea*. Interestingly, when *H. cunea* feed on the cyanogenic plant *Prunus serotina*, the caterpillar’s gut is maintained at a high pH, which prevents the formation of HCN from the cyanogenic glycosides in the food bolus [46]; thus, the caterpillar avoids being poisoned but does not gain benefit by sequestering a toxin. Several members of this subtribe have been reported as hosts of ichneumonoids [149], but the fact that most have not could easily reflect a paucity of rearing records as well as failure to publish parasitoids. 

Members of Lithosiini obligately feed on lichen and algae, from which they sequester phenolic compounds that are produced by the lichen fungal symbiont [121,152,153,154,155].

Members of Syntomini are generally medium-sized and highly aposematic. *Amata* is the largest and best-studied genus. Some species are highly polyphagous, while others are more specialized. The generalist *A. mogadorensis* is negatively affected by iridoid glycosides in *Plantago lanceolata* [85] and does not normally feed on toxic plants like many other Syntomini. Adults of some *Amata* species have a defensive pyrazine odor as is typical of ladybird beetles (Coccinellidae), viz., 3-isopropylypyrazine and 3-sec-butylypyrazine [156]. The source of these compounds in the unidentified, possibly undescribed, Australian *Amata* species that has been investigated is unknown, and in the case of this species, its feed on dead flowers and other plant litter. Some very limited experiments (due to a lack of available material) led Rothschild et al. [150] to suggest that the European *A. phegea* might contain a histamine-like substance.

Taxapad [149] includes only five ichneumonoid host–parasitoid records for *Amata*, all involving microgastrine braconids.

### 3.2. Nymphalidae

The classification of butterflies has undergone quite radical changes over the past 30 years or so, and an extensive phylogeny has just been published [157]. Nymphalidae is a large family, and it is now well established that the former separate families Acraeidae, Amathusiidae, Danaiidae, Heliconidae, Ithomidae, and Satyridae are all just derived clades within it. Most nymphalids are generally considered to be palatable, but several subfamilies and tribes include some of the most famous examples of aposematic coloration and Batesian and Müllerian mimicry.

Here, we deal specifically with five well-known groups of unpalatable species: Acraeini (Heliconiinae), Amathusiini (Morphinae), Danaini (Danainae), Heliconiini (Heliconiinae), Ithomini (Danainae), and Melitaeini (Nymphalinae). Most members of the other subfamilies and tribes are generally considered to be more or less palatable, although there are some exceptions. For example, *Junonia coenia* (Nymphalinae: Junoniini) feeds on plants rich in iridoid glycosides such as *Plantago,* its best-studied host, other Plantaginaceae, and Orobanchaceae, although they do not seem to sequester toxins to a particularly great extent [117], although catalpol was sequestered twice as efficiently as aucubin [158]. Another well-known example is *Taenaris*, a S. E. Asian genus in Morphinae (Amathusiini) whose caterpillars feed exclusively on cycads and sequester cycasin. It has only fairly recently been discovered that caterpillars of the South American blushing phantom butterfly, *Cithaerias pireta* (Satyrinae), feed on *Philodendron* (Araceae) [159]. Nearly all other satyrines feed on plant families such as grasses (Poaceae) that are not noted for containing significant quantities of secondary plant compounds, but may contain aroids such as long-chain alkyl resorcinols and their sugar analogues [160].

#### 3.2.1. Nymphalidae (Acraeini)

*Acraea* and relatives are a well-known tropical group of unpalatable nymphalids, in the past treated as a separate family, but now classified as a tribe of Heliconiinae. The sweet potato butterfly, *Acraea acerata*, is a serious pest of *Ipomoea* species (Convolvulaceae) in Africa, with sweet potato, *Ipomoea batatus*, being a major host plant.

Interestingly, American Acraeinae butterflies often feed on host plants rich in PAs (dehydropyrrolizidine alkaloids), notably various Asteraceae. However, these compounds are not normally sequestered for defense; instead, the larval and adult butterflies synthesize large amounts of the cyanogenic glucoside linamarin, which is what renders them toxic and protected [161].

#### 3.2.2. Nymphalidae (Danaini)

Danaines collectively feed on many plant species, mostly in Apocynaceae (e.g., *Asclepias*, *Cynanchum*, *Heterostemma*, *Hoya*, *Marsdenia*, *Metaplexis*, *Stephanotis,* and *Vincetoxicum*, to list just a few). However, some species utilize host plants in other families such as *Ficus* (Moraceae) and Menispermaceae, the former containing fouranocoumarins, and the latter being well known for its wide range of alkaloids. The best-studied in terms of toxicity are the N. American monarch butterflies, *Danaus plexippus* and *D. gilippus*, and the Afrotopical *D. chryssippus,* whose larvae feed on milkweeds (*Asclepias* spp.) [162,163].

Insect neuronal Na+/K+ ATPase is especially sensitive to cardiac glycosides, but unlike other insects, *Danaus plexippus* is insensitive to them. Holzinger et al. [164] amplified and cloned the DNA sequence encoding the putative 12-amino acid Na+/K+ ATPase binding site for the cardiac glycoside ouabain. They found that instead of having the amino acid asparagine at position 122, as in other insects that are sensitive to cardiac glycosides, *D. plexippus* has a histidine. Therefore, it seems likely that this evolutionary innovation explains the ouabain insensitivity of the monarch. That this amino acid substitution was important was confirmed by Holzinger and Wink [165], but they also showed that this was not the case in the related *D. gilippus,* nor in the ouabain-tolerant N. American arctiine *Syntomeida epilais* (see Section 3.1.2 above).

#### 3.2.3. Nymphalidae (Heliconiinae)

The tribe’s name comes from the type genus *Heliconius* (Figure 3F), of which there are many species in the Neotropics. Many of these are involved in complex mimicry rings, which have been the subject of a great deal of research on the evolution of unpalatablity and of mimicry.

Some Heliconiinae are able to synthesize cyanogenic compounds de novo, whereas others (e.g., *Boloria* (*Clossiana*) *euphrosyne*) possess 𝛽-cyanoalanine synthase, the same enzyme that zygaenids use to detoxify HCN, suggesting that this species avoids cyanide toxicity in this way [65].

#### 3.2.4. Nymphalidae (Danaiinae: ITHOMIINI)

Ithomiines are a neotropical group of butterflies with approximately 370 species and 40 or so genera. They are closely related to Danainae and some authors consider them as a tribe of the latter. Their caterpillars feed predominantly on various species of Solanaceae, but some genera feed on Echiteae vines (Apocynaceae) and a few feed on Gesneriaceae. From these plants, they sequester PAs [128,166]. 

#### 3.2.5. Nymphalidae (Nymphalinae: Melitaeini)

Melitaeini are generally regarded as unpalatable, though not particularly famously so. Collectively, they feed on 16 host plant families, of which 12 families have high levels of iridoid glycosides, especially members of Plantaginaceae [114].

The checker-spot butterfly, *Euphydryas editha* (Nymphalidae), feeds on various low-growing IG-containing plants [167], from which it sequesters IGs. Its common food plants include *Collinsia tinctoria* (Plantaginaceae), *Penstemon heterodoxus* (Plantaginaceae), *Plantago erecta* (Plantaginaceae), *Castilleja nana* (Orobanchaceae), *Orthocarpus densiflorus* (Orobanchaceae), although this plant usually lacks suitable oviposition sites close to the ground, and *Pedicularis densiflora* (Orobanchaceae) [167]. Wide individual and year-to-year variation in iridoid content was found in the congener *E. anicia* by [168]. Stamp [169] noted that defensive behaviors exhibited by caterpillars of the Baltimore checkerspot, *E. phaeton*, were really rather effective deterrents against parasitoidism too.

Only very few species of *Cotesia* (Braconidae: Microgastrinae) parasitize European Melitaeini. The larvae of these butterflies in Europe feed on various species of *Plantago* (Plantaginaceae), *Centaurea* (Asteraceae), *Linaria* (Scrophulariaceae), *Antirrhinum* (Scrophulariaceae), *Veronica* (Scrophulariaceae), *Digitalis* (Plantaginaceae), *Verbascum* (Scrophulariaceae), *Valeriana*, (Caprifoliaceae) *Melampyrum*, (Orobanchaceae) *Lonicera* (Caprifoliaceae), *Scabiosa* (Caprifoliaceae), *Succisa* (Caprifoliaceae), and *Gentiana* (Genianaceae) (M.R. Shaw, pers. comm.).

Parasitoids of the European Melitaeini are some of the best known of any butterflies [170,171,172,173,174]. These parasitoid complexes are rather atypical in that they are so strongly dominated by specialists, notably gregarious *Cotesia* species. In the Åland Islands in southwest Finland, the parasitoid complex of *Melitaea cinxia* has been studied for more than a decade. There are two specialist primary larval endoparasitoids, each with their own secondary parasitoids, and several generalist pupal parasitoids.

### 3.3. Papilionidae (Troidini and Parnassinae)

The spectacular (and much-prized by insect collectors) birdwing butterflies belong to the tribe Troidini. Their larvae mostly feed on Aristolochiaceae, notably *Aristolochia* species, but also some on *Thottea* (=*Apama*). Members of the Troidini are largely not parasitized [175]. The best-known genera are *Troides* (Figure 4A), *Trogonoptera*, *Ornithoptera*, *Battus*, *Atrophoneura*, *Losaria*, *Parides*, and *Pachliopta* (Figure 4B). Within the Americas, it appears that the choice of host plant species by species in the Troidini genera *Battus* and *Parides* is opportunistic; they will oviposit on whatever Aristolochiaceae they find within their range [55,176].

Parnassinae includes the collectable festoon (e.g., *Zerynthia* spp. (Zerynthiini)), false Apollo (e.g., *Archon* spp. (Luehdorfiini), and Apollo (e.g., *Parnassius* spp. (Parnassiini)) butterflies. The caterpillars of both Zerynthiini and Luehdorfiini are specialized *Aristolochia* feeders, whereas those of Parnasssiini feed on various Crassulaceae (e.g., *Sedum*, *Sempervivum*) and Papaveraceae (e.g., *Corydalis*).

Comparison of host plant phylogeny with that of Papilionidae [177] provides no evidence for co-cladogenesis, but rather that host plant toxins constitute a major barrier to colonization by papilionids.

### 3.4. Pieridae (Pierini)

The white, sulfur, and jezebel butterflies (Pieridae) are known to have approximately 1100 species worldwide. They are classified into four subfamilies and six tribes. Members of Pierini mostly feed on plants containing mustard oils as the primary defense chemicals [178].

Some other groups of Pieridae also feed on Capparidaceae. For example, the Pierini *Cepora* (Figure 4G) and Teracolini *Ixias* and are brightly colored, but also appear, at least at some times of year, to be Batesian mimics of *Delias* species [179], and it is not certain to what extent they may themselves be toxic.

### 3.5. Zygaenidae

Some Zygaenidae (burnet moths) sequester the cyanogenic glucosides linamarin and lotaustralin from their food plants (Fabaceae) [105]. *Zygaena* displays an exceptionally high content of cyanogenic glucosides within Zygaeninae (Figure 5G) [180] and this reflects the capacity to sequester and store the compounds in specialized storage chambers in the cuticle of Zygaeninae larvae (cuticular cavities) [181].

In a study conducted by Zikic et al. [182], several parasitoid wasp species were observed to emerge from *Zygaena filipendulae* larvae (Figure 5G) that had been collected feeding on grey elm trees, *Ulmus canescens* (Ulmaceae). These parasitoid wasps belonged to several families, including Braconidae (Microgastrinae), Ichneumonidae (Cryptinae, Mesochorinae), Eulophidae (Elasminae, Entedoninae), Eupelmidae (Eupelminae), and Chalcididae (Chalcidinae). Additionally, parasitoid tachinid flies (Diptera), specifically *Phryxe nemea*, were also identified in their study [182]. This is quite different from when *Z. filipendulae* feeds on herbaceous and usually cyanogenic clovers, trefoils, and vetches (Fabaceae).

Zygaenidae are normally attacked by only extremely specialized parasitoids [183,184].

### 3.6. Miscellaneous

Caterpillars, pupae, and adults of the European magpie moth, *Abraxas grossulariata* (Geometridae), are aposematic (Figure 6) and contain a cyanoglucoside, sarmentosin [185], as well as having histamine-like activity. The caterpillar feeds mostly on various Rosaceae, e.g., *Ribes* species grown in gardens, crab apple (*Malus pumila*), hawthorn (*Crataegus monogyna*), blackthorn (*Prunus spinosa*), and bird cherry (*P. padus*), but also the crassulaceans orpine (*Sedum telephium*) and pennywort (*Umbilicus rupestris*) (Crassulaceae) as well as a few other plants such as ling (*Caluna vulgaris*), Japanese spindle tree (*Euonymus japonicus*), ivy (*Hedera helix*), and elm (*Ulmus*). *A. grossulariata* caterpillars are known to be parasitized by 51 species of Ichneumonoidea including microgastrine braconids of the genera *Cotesia*, *Glyptapanteles,* and *Protapanteles* (see Microgastrinae in Section 4.1.3 below) as well as two records of the Euphorinae genus *Meteorus* [149].

Other aposematic geometrids include *Cystidia* species (Geometridae: Ennominae) [110], which feed on various Rosaceae and Celastraceae and are unpalatable to lizards, and *Arichanna*, [111] which sequesters grayanoid diterpenes from its food plant *Peris japonica* (Ericaceae) [186]. Konno et al. [187] report six species of hymenopterous parasitoids of *Cystidia couggaria*, but all of these belong to Chalcidoidea rather than Ichneumonoidea, viz., *Chouioia cunea* (Eulophidae), *Monodontomerus minor* (Torymidae), *Brachymeria lasus* (Chalcididae), *Brachymeria* sp. (Chalcididae), *Dibrachys cavus* (Pteromalidae), and *Eupelmus* sp. (Eupelmidae). All these moths have conspicuously colored larvae and pupae, which are formed in exposed positions, and the same is true for the toxic lymantriine *Ivela auripes* [188], whose caterpillars feed predominantly on *Cornus* species (Cornaceae). Yu et al. [149] record two species of Microgastrinae as parasitoids of these caterpillars, viz., *Cotesia melanoscela* and *Glyptapanteles liparidis*.

Members of the genus *Uresiphita* (Lepidoptera: Crambidae) appear to be specialists on various quinolizidine alkaloid-containing members of Fabaceae [189]. The N. American broom moth, *Uresiphita reversalis* [132], is cryptic as an adult, but its larvae are aposematic on their broom host plant, *Genista monspessulana* (Fabaceae: *Genista*). The host plant is rich in quinolizidine alkaloids, which the *U. reversalis* caterpillar sequesters into its cuticle. No parasitoid records appear to have been published for *U. reversalis*, but congeners are reportedly attacked by the euphorine braconid *Meteorus pulchricornis*.

The larvae of numerous moth and butterfly species are known to consume cycads [190], which are a rich source of secondary compounds including several that are carcinogenic and neurotoxic [191]. Some of their repertoire of secondary compounds appear to be the result of horizontal gene transfer from various micro-organisms, which confers insecticidal properties [192]. Many cycad specialists are warningly colored and sequester cycad toxins. Blue butterflies and hairstreaks (Lycaenidae) are not normally unpalatable, but some species that feed upon cycads are, and a subset of these display distinctive warning coloration [190]. Bowers and Larin [131] report a case of toxin sequestration by *Eumaeus atala*, a species from S.E. USA and the West Indies, which has a highly aposematic appearance (black, scarlet, and some white spots). Its native larval food plant is the cycad *Zamia integrifolia*, which is also known as the coontie palm.

## 4. The Parasitoids

Sequestered secondary plant compounds may affect a host insect’s immune system [39] and either their ability to encapsulate a parasitoid egg or first-instar larva. There are unfortunately very few data on exactly what negative effects plant metabolites might have on parasitoids [39,193]. 

It is not known how endoparasitoids of toxic caterpillars avoid poisoning themselves. It might be presumed that as with other animals, dietary exposure to toxic compounds might induce the production of P450 detoxication enzymes [194]. If the parasitoid has not evolved to sequester toxins from its host, and this mostly seems to be the case, then they presumably detoxify consumed toxins or minimize ingesting them, perhaps by avoiding feeding on the most toxin-laden toxic host tissues. Although some larger tropical ichneumonoids are very brightly colored (especially some Braconidae) and are possibly models of homochromatic complexes [195,196,197], there is scant evidence that most are unpalatable due to chemicals, and if they are, these compounds appear to be produced by the wasps themselves [51].

### 4.1. Braconidae

There is now a well-supported phylogenetic hypothesis for the relationship between most braconid subfamilies [198,199]. The major groups whose members predominantly parasitize exposed or weakly concealed Lepidoptera caterpillars are (a) Rogadinae, (b) all members of the microgastroid complex of subfamilies, (c) the macrocentroid sub-complex, and (d) most members of the eurphorine tribe Meteorini.

All the other subfamilies examined here belong to the non-cyclostome lineage [51] and display very typical koinobiont syndromes of small eggs and pro-ovigeny. The cyclostome subfamily Rogadinae represents a second, independent evolution of koinobiosis within Braconidae, and they are atypical koinobionts in that they are synovigenic and lay relatively large yolky eggs [200].

#### 4.1.1. Euphorinae (Meteorini)

This is a very large group of braconid wasps, often treated as a separate subfamily (Meteorinae). They are most likely the sister group of the remaining Euphorinae (which have diverse and mostly very different biologies) [51]. The tribe comprises the vast cosmopolitan genus *Meteorus* plus the relatively less common genus *Zele*. Most species are parasitoids of Lepidoptera, but a few are known from Coleoptera. Collectively, they parasitize members of many different moth and butterfly families. 

*Meteorus pulchricornis* is of note because out of the 84 host records representing 20 different Lepidopteran families in [149], 3 are Arctiinae (*Arctia festiva*, *Hyphantria cunea,* and *Spilarctia obliqua*). This species is broadly dispersed around the world and is easy to culture in the laboratory, which should make it a good candidate for investigating how koinobiont endoparasitoids cope with toxic hosts. Of course, it might be that the individual arctiids it was reared from were not especially rich in sequestered toxins. Another polyphagous species is *Meteorus hyphantriae*, a parasitoid of the fall webworm, *H. cunea*, a very destructive arctiine pest of a number of ornamental trees, shrubs, and agricultural crops. However, this particular arctiine does not sequester plant secondary compounds and instead appears very able to detoxify them in its gut [201].

Shaw and Jones [202] described a new species of *Meteorus* that is a parasitoid of the toxic butterfly *Pteronymia zerlina* (Nymphalidae: Danainae, Ithomii) in Ecuador. The butterfly’s larvae are solitary and contrastingly colored, and feed on various species of *Solanum*; therefore, they are presumed to be toxic. However, the unpalatable adult ithomiines obtain their protective PAs through adult flower feeding and not from sequestered food plant compounds [203].

#### 4.1.2. The Macrocentroid Complex

This group of subfamilies comprises the Amicrocentrinae (one Afrotropical genus parasitizing endophagous caterpillars), Charmontiinae, Homolobinae, Macrocentrinae, Microtypinae, Orgilinae, and Xiphozelinae. They are almost exclusively parasitoids of exposed or weakly concealed, externally feeding caterpillars, but include leaf rollers and tiers.

##### Charmontiinae

Charmontiinae comprises the cosmopolitan genus *Charmon* and another monotypic one from Chile. *Charmon* is thus far known from 10 species [204], but is notable in that one Holarctic species, *C. extensor*, has a particularly large host range, having been recorded from 12 families of Lepidoptera. Whether *C. extensor* is really just a single highly polyphagous species or perhaps a complex of very similar more specialized species has not yet been tested, although there is some evidence that European and North American specimens going under this name are likely to be different [204].

##### Homolobinae

Homolobinae is another small cosmopolitan group with 67 described species in 3 genera. They are mainly parasitoids of Geometridae, Noctuidae, Erebidae (Lymantriinae), and Lasiocampidae.

##### Macrocentrinae

Macrocentrinae is a relatively large subfamily with much undescribed diversity in the tropics. There are approximately 250 described species in eight genera. Their hosts include concealed and exposed caterpillars, predominantly Erebidae, Gelechidae, Noctuidae, Oecophoridae, Pyralidae, and Sesiidae. Many species are reported to have wide host ranges.

##### Microtypinae

Microtypinae is a small but virtually cosmopolitan group (except Australia) and comprises 23 known species classified into 3 genera, but its biology is only known for Holarctic species, which include parasitoids of concealed larvae of ‘microlepidoptera’ of the families Pyralidae, Gelechiidae, Tortricidae, and Yponomeutidae.

##### Orgilinae

Orgilinae comprises 362 described species in 13 genera. They are generally best represented in the tropics. Their known hosts belong mainly to Coleophoridae, Gelechiidae, Gracillariidae, Oecophoridae, Pyralidae, Crambidae, and Tortricidae. 

##### Xiphozelinae

Xiphozelinae is restricted to S.E. Asia and adjacent areas, and comprises six species in two genera [205]. The only host record is of a *Xiphozele* species reared from the non-toxic erebid *Bastilla simillima* (as *Dysgonia*, as *Ophiusa*).

#### 4.1.3. Microgastroids (Cardiochilinae, Cheloninae, and Microgastrinae)

Microgastroids include some of the most familiar of parasitoid wasps to the amateur entomologist and gardener, who will frequently encounter clusters of their cocoons attached to the remains of their lepidopteran host caterpillar. In the analyses presented in Section 5, we include just two subfamilies; the remaining few are parasitoids of endophytic leaf-miners (Miracinae) or very poorly known (Dirrhopinae, Khoikhoinae, and Mendeselinae). All studied members of this complex possess polydnaviruses [206,207]. The particles are created in huge numbers in swollen structures, called the calyx glands, at the posterior ends of the lateral oviducts [51]. Ancestrally, these were independent insect viruses, but their genomes became integrated into those of the parasitoid, and now they serve as a delivery mechanism for genes that will be expressed only inside host cells, to the benefit of the parasitoid. It is possible that this evolutionary adaptation may confer a significant advantage for successfully parasitizing toxin caterpillar hosts, but this aspect does not seem to have been investigated. The main reason for this is that most experimental work concerns systems that are either easy to culture in the laboratory or are important in terms of ago-forestry, and these do not usually involve toxic hosts. Furthermore, while Microgastrinae collectively have a very large number of associations with toxic hosts, the biologically very similar Cardiochilinae are not remarkable in this respect (see Section 5.1 below).

##### Microgastrinae

By far the largest number of host records involve the huge subfamily Microgastrinae. Some 3000 species have been described [208], but the majority are undescribed, and it has been estimated that there may be as many as 20–40,000 [209]. This subfamily is of great significance for the biological control of agricultural and forestry lepidopterous pests globally, owing to their extensive diversity, wide distribution in terrestrial habitats, and their exclusive parasitization of larval Lepidoptera from almost all families within the taxon Eulepidoptera [210,211]. These wasps are all koinobiont endoparasitoids and exhibit a remarkable ability to parasitize nearly the entire taxonomic and biological spectrum of Lepidoptera, with only the four most basal superfamilies being probable exceptions [211].

In general, the presence of microgastrine species in toxic host groups is relatively limited and the majority of them belong to a limited subset of genera, viz., *Cotesia*, *Glyptapanteles*, and *Protapanteles* [149], all of which may be placed in the informal "*Cotesia* group" of genera, although this is likely not a monophyletic group. Reports of members of certain other genera such as *Apanteles*, *Parapanteles,* and *Hypomicrogaster* being reared from papilionid and toxic nymphalid hosts are limited and have received little significant attention. Members of the ‘*Cotesia* group’ are particularly associated with "macrolepidoptera" hosts [212]. 

*Cotesia acerbiae* (Microgastrinae) and *Meteorus acerbiavorus* are two gregarious parasitoids of *Acerbia alpina* (Erebidae: Arctiinae) larvae [213,214] that have a varied diet, consuming plants from three different families, namely Asteraceae (*Taraxacum officinale*), Ericaceae (*Vaccinium*), and Salicaceae (*Salix herbacea*). When the caterpillars feed on *Taraxacum officinale*, they are likely exposed to levels of PAs that would typically deter parasitoids. However, both parasitoids have successfully adapted to this host, apparently as specialists, despite the potential presence of PAs. Since the host may have been feeding on non-PA-containing plants much of the time, this might have facilitated successful parasitism by *C. acerbiae*.

*Cotesia congregata* is a gregarious microgastrine that is largely a specialist on various hawkmoth caterpillars (Sphingidae), although it also attacks a few semi-permissive noctuids. It is best known as a biological control agent against the pest tobacco hornworm, *Manduca sexta*, especially in the S.E. USA. Its success in completing development to adulthood has been shown to be negatively related to the concentration of nicotine alkaloids in its diet [122]. It also feeds on the catapol (iridoid glycoside)-sequestering catalpa sphinx moth caterpillar, which has been shown to accumulate very small amounts of the toxin from its host [15], and small amounts of catapol were also detected in the larval cocoons.

The only example we can find of an ichneumonoid parasitoid sequestering toxins from its host involves an unidentified species of *Microplitis* from New Zealand [215]. Its host is the *Nyctemera annulata* (Erebidae: Arctiinae, Arctiini, Nyctemerina), which feeds on a range of introduced species of *Senecio* (Asteraceae) from which it sequesters PAs. The common name of the moth is the magpie moth, not to be confused with the Eurasian geometriid (see Section 3.6 above).

##### Cardiochilinae

Cardiochilinae are similar in many respects to Microgastrinae but are generally rather larger-bodied, and they are almost entirely restricted to warmer parts of the world. It is a far smaller subfamily than Microgastrinae, with only some 220 species known, classified among 18 genera. *Toxoneuron nigriceps* is an important parasitoid of the tobacco budworm moth, *Heliothis virescens* (Noctuidae), and so has been the subject of a great deal of research. However, none of its host records, nor those for any other members of the subfamily, that are on Taxapad [149] involve members of any of the traditionally unpalatable groups of Lepidoptera.

##### Cheloninae

Cheloninae is a sister group to the remainder of the traditional Lepidoptera-parasitizing microgastroids. Unlike the others, all chelonines are egg–larval parasitoids, and virtually all are solitary. Out of more than 600 host–parasitoid records for Cheloninae on Taxapad [149], only 3 are from unpalatable hosts, and all of these for Arctiinae.

#### 4.1.4. Rogadinae

This is a moderately diverse subfamily that are exclusively parasitoids of caterpillars [51]. We include here members of the tribes Aleiodini, Rogadini, and Yeliconini. The entirely New World Stiropiini is a small group that are all parasitoids of leaf miners (Lyonetidae and Gracillariidae). Nothing is known of the biology of Betylobraconini. In terms of described species, the subfamily is dominated by the genus *Aleiodes*, followed by *Triraphis* (Rogadini). Many genera within Rogadinae (Rogadini) are known to parasitize members of Zygaenoidea [184,216], including some Zygaenidae. This superfamily also includes Zygaenidae, Dalceridae, Epipyropidae, Heterogynidae, Himantopteridae, Lacturidae, Limacodidae, Megalopygidae, and a few other less well-known families. At least some Heterogynidae, Limacodidae, and Megalopygidae as well as Zygaenidae include species capable of cyanogenesis [65]. However, the West Palaearctic *A. assimilis* is the only species of the vast cosmopolitan genus *Aleiodes* known to attack them (M. R. Shaw, unpublished in Quicke et al. [184]). 

### 4.2. Ichneumonidae

Phylogenetic relationships among the ichneumonid subfamilies, although not fully resolved, show clearly that there are multiple origins of koinobiont endoparasitism of Lepidoptera caterpillars [217,218]. The great majority of species with this biology belong to a large group of subfamilies referred to as ophioniformes. The subfamilies relevant here are Anomaloninae, Banchinae, Campopleginae, Cremastinae, and Ophioninae, although most of these also include at least one parasitoid of beetles. Ichneumoniformes include a mix of idiobiont ectoparasitoids of various insect groups, but all members of the large subfamily Ichneumoninae are endoparasitoids of lepidopteran hosts, some being idiobiont pupal parasitoids and others koinobiont, often, but not always, attacking later-instar host larvae, but completing feeding after the host has pupated.

#### 4.2.1. Anomaloninae

A cosmopolitan subfamily of medium to large wasps that are solitary koinobiont endoparasitoids. The monotypic Anomalonini are parasitoids of concealed Coleoptera, and the large tribe Gravenhorstiini collectively attack a wide range of lepidopteran families. The most detailed investigation of their biology is still that of Tothill more than 100 years ago [219]. Oviposition is into the larval stage of the host, but development does not proceed past the first larval instar until after the host has pupated.

Anomaloninae appear to be particularly capable of developing on toxic host groups. The Taxapad dataset [149] includes 49 separate records: one from Againinae and Arctiinae (especially Arctiini: Phaegopterina, but including one from Ctenuchina), seven from Parnassiinae, ten from Pierinae, and five from Zygaenidae (all from *Zygaena*).

#### 4.2.2. Banchinae

A common and large subfamily of predominantly solitary, koinobiont endoparasitoids of Lepidoptera. Members of Banchini have short ovipositors and attack exposed hosts, whereas Atrophini and Glyptinae generally have long ovipositors and attack weakly concealed hosts such as leaf tiers and leaf-rollers, especially Tortricidae but also of numerous other families [51]. Some Atrophini also have short ovipositors and include species that attack Arctiinae. 

Several species of Arctiinae (Arctiini and Lithosiini) are reportedly attacked by various species of *Exetastes* (Banchini), including *Exetastes illusor* from *Tyria jacobaeae* (Figure 2). Additionally, there are a few records of various genera attacking Zygaenidae. There are no records of hosts in Meltaeini or Papilionidae.

It has recently been discovered that at least some Banchinae possess polydnaviruses, but since they are only distantly related to Campopleginae, this clearly reflects an independent acquisition, even though the ancestral free-‘living’ virus may have been the same [220].

#### 4.2.3. Campopleginae

This is a vast but taxonomically difficult group of parasitoids with many species investigated or utilized in biological control or IPM. Most species are parasitoids of Lepidoptera caterpillars, although a few genera are specialized on various coleopteran hosts [51]. Oviposition is normally into early-instar hosts, and, depending on species, pupation can be external (the normal mode) or within the caterpillar’s remains. Most that have been studied have associated polydnaviruses, although the first to be investigated in any detail, *Venturia canescens*, instead produces in its calyx glands protective particles that lack viral DNA.

*Campoletis sonorensis* is an important parasitoid in the control of the tobacco budworm caterpillar, *Heliothis virescens*. The pyridine alkaloid nicotine was shown to be nearly four times more toxic to parasitized versus unparasitized horn worms, thus complicating the trade-offs between nicotine concentration and numbers of hosts and parasitoids reaching the next generation [221]. Gunasena et al. [221] also showed that nicotine concentration was negatively correlated with the successful egression of fully fed *C. sonorensis* larvae from the host.

*Hyposoter exiguae* is a well-known solitary generalist endoparasitoid important in the control of many caterpillar pests such as the highly polyphagous corn earworm, *Heliothis zea* (Noctuidae). When *H. zea* caterpillars feed on tomato, they sequester the steroidal glycoalkaloid α-tomatine, and this has been shown to have a negative effect on the development of *Hyp. Exiguae,* which is used as a control agent [133]. Because of this, there is a conflict between use of the ichneumonid to control the pest and breeding tomatoes for higher resistance against herbivores. It should be noted that α-tomatine is present in tomato plant stems and leaves but only in minute, non-harmful concentrations in the fruit.

#### 4.2.4. Cremastinae

Cremastinae is a medium-sized subfamily largely restricted to warmer parts of the world. Their hosts are mostly weakly concealed exophagous Lepidoptera such as leaf-rollers and tiers, but some also attack beetle larvae in similar micro-habitats [51]. Their main lepidopteran hosts are Tortricidae, Pyralidae, Noctuidae, and Gelechiidae. Only one species, the Old World tropical *Trathala noxiosa*, is recorded from an arctiine, *Spilosoma* (=*Spilarctia*; =*Nebarctia*) *obliqua* [149,222]. Given that the host caterpillars are exposed at all times, the possibility that this record is incorrect must be considered, although most records by [222] are reliable. Therefore, this subfamily seems not to utilize toxic hosts, although this might also be related to the fact that the hosts are concealed and that excludes all the well-know aposematic groups.

#### 4.2.5. Ichneumoninae

Ichneumoninae are exclusively Lepidoptera endoparasitoids, but most are idiobionts attacking the host pupa, while some are technically koinobionts, attacking a final-instar caterpillar shortly before it pupates, often after the host has ceased feeding and is searching for a pupation site. In this way, they differ rather a lot from ophioniformes, many of which attack early-instar larvae but do not complete development until the host is much more developed.

The ichneumonine *Trogus pennator* is one of the best-studied of the parasitoids of Papilionidae larvae in the genera *Eurytides* and *Papilio*, but under normal circumstances, it does not attack the unpalatable swallowtail *Battus philenor,* even when its larvae co-occur in an area with those of other acceptable hosts [53]. The wasp is attracted by the caterpillar frass of *B. philenor*, but the female rejects their caterpillars upon antennation. Despite this reluctance, [53] was occasionally able to fool the *Trogus* by presenting *B. philenor* larvae along with acceptable hosts and their frass at close quarters. On these occasions, it was found that the parasitoid larvae were unable to develop due to some aspect of the host’s physiology or chemistry, likely the latter.

#### 4.2.6. Metopiinae

A medium-sized, cosmopolitan subfamily that are solitary koinobiont endoparasitoids of exposed and weakly concealed hosts. They emerge from the host pupa. What little is known about their developmental biology is based largely on just a few European species [51]. Oviposition is into a caterpillar stage, but development is only completed after the host has pupated. 

Several *Metopius* species have been reported from Arctiinae [149]: four collectively from Spilosomini (*Watsonarctia* and *Spilosoma*) and two from Arctiini (*Rhyparioides*). An undescribed *Corsoncus* species has been reared from the toxic arctiine *Utetheisa ornatrix* [14].

#### 4.2.7. Ophioninae

This subfamily includes a few common genera whose species are often moderately large, conspicuous, yellow-brown nocturnal visitors to lights and so may come to the attention of non-entomologists rather more often than many other groups of parasitoid wasps. With a single known exception [51], they are solitary koinobiont endoparasitoids of caterpillars. Despite their abundance, rather little is known about their developmental biology.

The majority of host records are from larger-bodied moths, e.g., Noctuidae, Lasiocampidae, Erebidae, Sphingidae, Saturniidae. Only two families of toxic hosts are attacked, most records are from Arctiinae, and most of these belong to the relatively weakly chemically defended subtribe Spilosomina, which includes *Hyphantria*, *Hyperconte*, *Pyrrharctia*, *Rhodogastria*, *Spilarctia* (Figure 1F), and *Spilosoma*. The record of an *Ophion* from the Eurasian garden tiger moth, *Arctia caja,* is wrong (G. Broad, pers. comm.), and the record of *Enicospilus repentinus* from *Zerynthia rumina* (Papilionidae: Parnassiinae) is definitely wrong because ophionines require subterranean or otherwise enclosed pupation sites host whereas *Zerynthia* pupate exposed on a plant stem (M.R. Shaw, pers comm.).

## 5. Analysis of Taxapad 2016 Ichneumonoidea Host Record Data

The Taxapad database, initially released as a CD and subsequently accessible as a USB drive or a partial web product (currently offline), has been the standard reference catalogue of Ichneumonoidea, featuring approximately 350,000 names, for almost 15 years [149]. It should be noted that the database is fundamentally a compilation of all published information, including incorrect information. Nonetheless, Taxapad offers a remarkable resource containing abundant information on the taxonomy, distribution, hosts and host plants, morphology, and more of Ichneumonoidea that is easily organized and analyzed. Consequently, researchers worldwide frequently consult it, and it has been adopted and employed (unfortunately, without critical examination) in many other databases, websites, and publications that pertain to Ichneumonoidea. Nevertheless, it is imperative to authenticate and verify the hosting information in Taxapad to establish its precision and dependability.

Our current understanding of host–parasitoid relationships is limited and incomplete, and a considerable proportion of published host records have been found to be incorrect. These errors may result from a range of factors, such as misidentification of either the host or the parasitoid, incomplete or insufficient data, or lack of knowledge about the biology and behavior of the organisms involved. Furthermore, new host–parasitoid associations are continually being discovered, and the true extent of parasitoid diversity and host range remains largely unknown. To address these knowledge gaps, researchers have employed a variety of approaches, including field surveys, laboratory experiments, and molecular analyses. However, even with these tools at their disposal, many host–parasitoid relationships still elude detection. In some cases, this may be due to the cryptic nature of some parasitoids, which can make them difficult to observe or collect. Additionally, many parasitoids are highly host-specific, making them less likely to be encountered in generalist surveys.

### 5.1. Results from Taxapad 2016 Braconidae Host Records

For Braconidae, we extracted data for braconid subfamilies whose members are entirely or predominantly koinobiont endoparasitoids of mostly exophytic, folivorous caterpillars, specifically Agathidinae, Cardiochilinae, Charmontinae, Cheloninae, Euphorinae (Meteorini), Homolobinae, Macrocentrinae, Microgastrinae, Microtypinae, Orgilinae, Rogadinae, and Sigalphinae.

The overall results from the Taxapad 2016 dataset are summarized in Table 2. It can be seen that for slightly more than half of the Lepidoptera-associated koinobiont endoparasitoid subfamilies, there were no host records from the nine target groups of generally toxic Lepidoptera. The numbers of individual published host records vary widely between braconid subfamilies from twenty or fewer for Meteorideinae, Sigalphinae, and Microtypinae to over one thousand for both Meteorini and Microgastrinae. Indeed, the Taxapad dataset includes many more host records for Microgastrinae than for all the other koinobiont exophytic Lepidoptera-parasitizing braconids combined (6841 vs. 4620).

Given that even for those groups that are recorded from toxic hosts, the proportion of such records is, in most cases, less than 0.01, we cannot say that most of the braconid subfamilies are unable successfully to parasitize toxic hosts. That there are more than 500 host records each for Agathidinae and Cheloninae yet none from the targeted toxic host taxa is suggestive that members of these two subfamilies are less able to cope with toxic hosts than some other subfamilies. The strong exception is the Rogadinae.

Yu et al. [149] include information on 6979 rearing records from lepidopteran hosts (Table 2). When the number of unique parasitoid host records in a given lepidopteran family (or group) is plotted against the number of host species recorded in each group (Figure 7), it is apparent that collectively, there are relatively few species of microgastrine recorded from the toxic host groups.

We then conducted a separate analysis for the other subfamilies combined (Figure 8) to see whether the Microgastrinae result was likely to be representative. Whilst collectively, at least a few species of Microgastrinae have been reported as parasitizing all nine of our largely toxic groups of butterflies and moths, all the rest of the considered Braconidae combined had only been recorded from five toxic host groups, and three by just a single record.

**Table 2 toxins-15-00424-t002:** Number of unique braconid wasp host associations for taxa specialized on Lepidopteran hosts (Source: data from Yu et al. [149]).

Subfamily (Total Number of Published Host Records)	Number of Associations with Palatable Host Groups	Number of Associations with Unpalatable Host Groups	Proportion of Associations Involving Unpalatable Hosts
Agathidinae (658)	570	0	0
Cardiochilinae (81)	57	0	0
Charmontinae (90)	85	1	0.0012
Cheloninae (652)	556	0	0
Euphorinae (Meteorini) (1247)	1125	7	0.006
Homolobinae (137)	123	0	0
Macrocentrinae (696)	588	1	0.0017
Meteorideinae (13)	12	0	0
Microgastrinae (6979)	5823	55	0.009
Microtypinae (20)	19	0	0
Orgilinae (240)	210	2	0.009
Rogadinae (768)	695	18	0.026
Sigalphinae (19)	18	0	0

### 5.2. Results from Taxapad 2016 Ichneumonidae Host Records

Numerous subfamilies of Ichneumonidae parasitize caterpillars, and although there are nearly always some exceptions, most subfamilies/tribes of koinobionts specialize on a given insect order. Among the ‘ophioniformes’, exophytic Lepidoptera caterpillars are the predominant hosts for Anomaloninae, Banchinae, Campopleginae, Cremastinae, Metopiinae, and Ophioninae, as well as for the tribes Phytodietini and Oedemopsini of Tryphoninae. However, the Lepidoptera parasitizing tryphonines are koinobiont ectoparasitoids and therefore potentially not directly comparable with the other ophioniformes. Among the ‘pimpliformes’ and ‘ichneumoniformes’, there is far greater intra-subfamilial variation in host groups (many are parasitoids of Coleoptera or Diptera, and many also attack xylophagous hosts). The main Lepidoptera specialist subfamily among these is Ichneumoninae (see Section 4.2.5 above). We did not consider the ophioniformes subfamily Mesochorinae, which are hyperparasitoids rather than primary parasitoids of Lepidoptera. As with Braconidae, we only considered records from Lepidoptera, thus excluding the exceptions. The data are summarized in Table 3.

The Lepidoptera-parasitizing ‘ophioniformes’ are generally far less able to attack toxic hosts caterpillars than non-toxic ones (Figure 9). Figure 10 unmistakably illustrates the notable paucity of ichneumonine species recorded from the toxic host groups. Taxapad 2016 [149] includes records from only eight of the toxic groups considered here, and two of these are based on a single record each, and therefore need confirmation.

### 5.3. Comparison of Braconidae and Ichneumonidae 

Figure 11 presents the number of unique records of parasitoidism of toxic host Lepidoptera groups for the major subfamilies and groups of braconids and ichneumonids. There is at least one record of a member of each family attacking at least one member of the ten specified groups of generally toxic hosts. In the case of Braconidae, the host records are dominated by the single subfamily Microgastrinae, while the distribution of records from across the specified subfamilies of Ichneumonidae is somewhat more even. 

Nearly all the parasitoid wasp subfamilies considered could parasitize members of Arctiinae and Zygaenidae, and a substantial proportion could also attack members of Pierinae and Melitaeinae (Nymphalidae) (Figure 11).

## 6. Discussion

Price [223] commented that the absence of sequestered toxins in generalized herbivores not only shapes their own ecological dynamics but also has implications for the evolution of their natural enemies, particularly generalist parasitoids. Generalized herbivores, by lacking the ability to accumulate and store toxins from their host plants, offer a relatively toxin-free resource for parasitoids to exploit. This absence of sequestered toxins provides an opportunity for the emergence and persistence of generalist parasitoids, as they are not limited by specialized adaptations required to tolerate or detoxify specific host toxins. The evolution of generalist parasitoids in the absence of sequestered toxins in their herbivorous hosts can be attributed to several factors. By targeting generalist herbivores, generalist parasitoids can access a diverse and potentially more abundant resource base, enhancing their fitness and reproductive success.

It is now widely accepted that Ehrlich and Raven’s [224] contention that secondary plant compounds act as ‘barriers’ to insect feeding and colonization as host plants is generally true [177]. Whilst it is well known that toxicity of insect hosts provides protection against parasitoids [225], the magnitude of the protection across Lepidoptera has not previously been assessed.

We must bear in mind that with the combination of our sparse secure knowledge of latitudinal trends in both ichneumonoid families [28] and the even sparser and more biased published records of host–parasitoid relationships, there are limitations on what conclusions might be drawn. Nevertheless, here, we show that the number of ichneumonoid host–parasitoid associations involving general toxic groups of Lepidoptera are far lower than the average for most other utilized lepidopteran host families (Figure 7, Figure 8, Figure 9 and Figure 10).

The data analyzed suggest that among the toxic Lepidoptera, members of Arctiinae and Zygaenidae are the easiest for ichneumonoids to incorporate into their host ranges and eventually specialize on. Here, we discuss possible reasons for this. In contrast, Troidini and Parnassiinae, which sequester aristolochid acids, seem especially well protected from parasitoid attack, although these are not very species-rich groups compared to the others. Heliconiinae are also seldom utilized as hosts, with substantial numbers of parasitoid associations being restricted to Microgastrinae and Ichneumoninae. 

Unlike most of the other groups of Lepidoptera that sequester toxic secondary plant compounds, many arctiines do so only facultatively, and may even adjust their foodplant choice depending on whether or not they have been parasitized. Therefore, there is likely to be a proportion of individuals in the population that have low or even zero levels of sequestered plant toxins. Even though they may produce some of their own defense molecules (e.g., histamine), these individuals would likely be far easier for a generalist parasitoid to develop on successfully from time to time. In the case of some more specialist Arctiinae, e.g., the speckled footman moths, genus *Utethesia*, there may still be a great deal of inter-individual variation in the concentration and precise chemical structures of the sequestered plant toxins [226].

The sequestered or de novo-synthesized protective cyanogenic compounds (CGs) of zygaenid caterpillars are stored in specialized cavities in the cuticle [47,180,181,227]. Chemical analyses of *Zygaena filipendulae* tissues have shown that the CG concentrations were highest in the hemolymph and epidermis/cuticle but low in, for example, the fat body. Therefore, selective feeding by parasitoid larvae on host tissues containing lower levels of CGs could enable them to avoid toxicity caused by CGs. 

Although other evolutionary aspects could, in theory, be examined statistically, the current lack of accurate information on the total species diversities of all the parasitoid groups and the possibility of misidentifications based solely on morphology make further statistical exploration of the Taxapad data [149] premature.

## 7. Conclusions

We have shown that the classic exemplar toxic Lepidoptera groups are largely protected against ichneumonoid parasitoids. Their toxins pose a barrier to parasitoid species incorporating them into their host range. For all the groups of toxic hosts explored here, the number of unique ichneumonoid–host records was fewer than for the vast majority of non-toxic host groups (Figure 7, Figure 8, Figure 9 and Figure 10).

Hymenopteran parasitoids belonging to the microgastroid assemblage of braconid subfamilies (e.g., Cardiochilinae, Cheloninae, and Microgastrinae) and to the ichneumonid subfamilies Campopleginae and Banchinae (but not other members of the ‘ophioniformes’) produce polydnavirus particles that are co-injected into hosts along with their eggs and venom components [206,207]. The particles are created in very high numbers in the swollen structures at the posterior ends of the lateral oviducts called the calyx glands [51]. Ancestrally, these were independent insect viruses, but their genomes became integrated into those of the parasitoid, and now they serve as a delivery mechanism for genes that will be expressed only inside host cells, to the benefit of the parasitoid. However, it is not known whether, in instances of these wasps attacking toxic caterpillars, the polydnaviruses interfere with the host’s toxin production or sequestration mechanisms. The main reason being that most experimental work concerns systems that are either easy to culture in the laboratory or are important in terms of agro-forestry, and these do not usually involve toxic hosts.

A great deal is still unknown about how specialist parasitoids are able to actually survive in their toxic host caterpillars/pupae. We postulate here that at least in the case of some endoparasitoids, their larvae might selectively avoid feeding on host tissues that are particularly rich in sequestered toxins. In the case of cyanogenesis in Zygaenidae caterpillars, the tissues with the highest CG concentrations are the hemolymph and epidermis, with far lower amounts in other tissues. Therefore, it would be most interesting to discover what tissues the developing parasitoid larvae consume. Much more research could valuably be focused on the quantitative analysis of toxins and their catabolic products in different host tissues as well as in the developing parasitoids. Some combination of careful dissection and modern imaging techniques might enable the specific host tissues consumed by developing parasitoid larvae to be determined.

Further research can delve into the molecular mechanisms underlying toxin sequestration, the specific adaptations of parasitoid wasps to cope with toxic hosts, and the co-evolutionary dynamics between these organisms. The substantial parasitoid rearing program led by Daniel Janzen and Winnie Halwachs in Costa Rica will provide a great deal more evidence about host (and often tritrophic) interactions in this exemplar tropical country [Quicke et al., in prep.].

Given that some mechanisms have already been determined at the molecular level for herbivorous insects feeding on toxic host plants, there is a great need for biochemical investigation of detoxification mechanisms in the larvae of their parasitoids. 

## 8. Materials and Methods

Data on published host–parasitoid records for braconid and ichneumonid wasps were obtained from the Taxapad 2016 Ichneumonoidea database [149]. The search encompassed the period 1792 to 2015 for the family Braconidae and 1763 to 2016 for the family Ichneumonidae. However, the focus was placed on the most recent articles, which were carefully reviewed and evaluated based on specific inclusion criteria. These criteria included (a) access to the complete article content, (b) extraction and organization of metadata, (c) selection of caterpillar hosts, (d) investigation of their toxicity, and (e) consideration of prospective study designs. All data parsing and calculations were performed using the statistical computing language R [228].

## Figures and Tables

**Figure 1 toxins-15-00424-f001:**
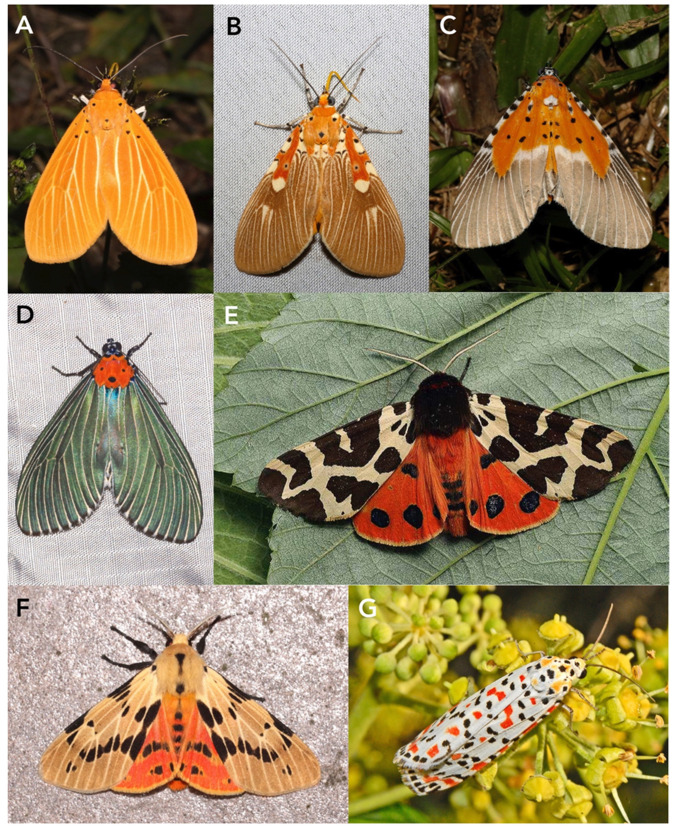
Exemplar adults of toxic species aganaine and arctiine moths. (**A**) *Asota egans* (Erebidae: Aganinae) from Thailand; (**B**) *Asota ficus* (Erebidae: Aganinae) from Thailand; (**C**) *Peridrome subfascia* (Erebidae: Aganinae) from Thailand; (**D**) *Neochera dominia* (Erebidae: Aganinae) from Thailand; (**A**–**D** photographs © Antonio Giudici, reproduced with permission); (**E**) garden tiger moth, *Arctia caja* (Erebidae: Arctiinae) from France (photograph by Jean-Pierre Hamon reproduced under terms of GNU Free Documentation License); (**F**) *Spilarctia* sp. Erebidae: Arctiinae, from Thailand (photograph © Antonio Giudici, reproduced with permission); (**G**) crimson-speckled footman moth, *Utetheisa pulchella*, (Erebidae: Arctiinae) from Italy (reproduced under terms of Creative Commons Attribution License 4.0, credit Hectonichus).

**Figure 2 toxins-15-00424-f002:**
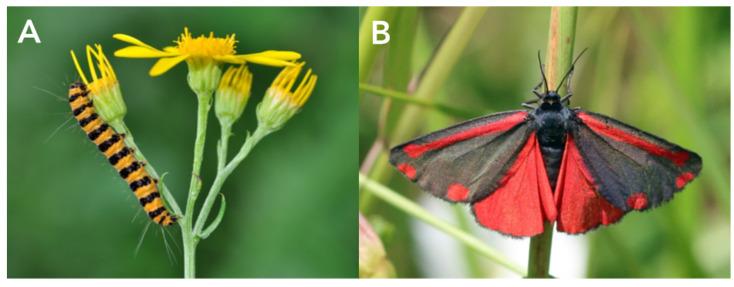
Cinnabar moth, *Tyria jacobaeae* (Erebidae: Arctiinae). (**A**) Caterpillar on its host plant, ragwort, *Jacobaea vulgaris* (reproduced under terms of Creative Commons Attribution-Share Alike 3.0 license, credit Quartl); (**B**) adult (reproduced under terms of Creative Commons Attribution-Share Alike 4.0 International license, credit Charles J. Sharp, Sharp Photography).

**Figure 3 toxins-15-00424-f003:**
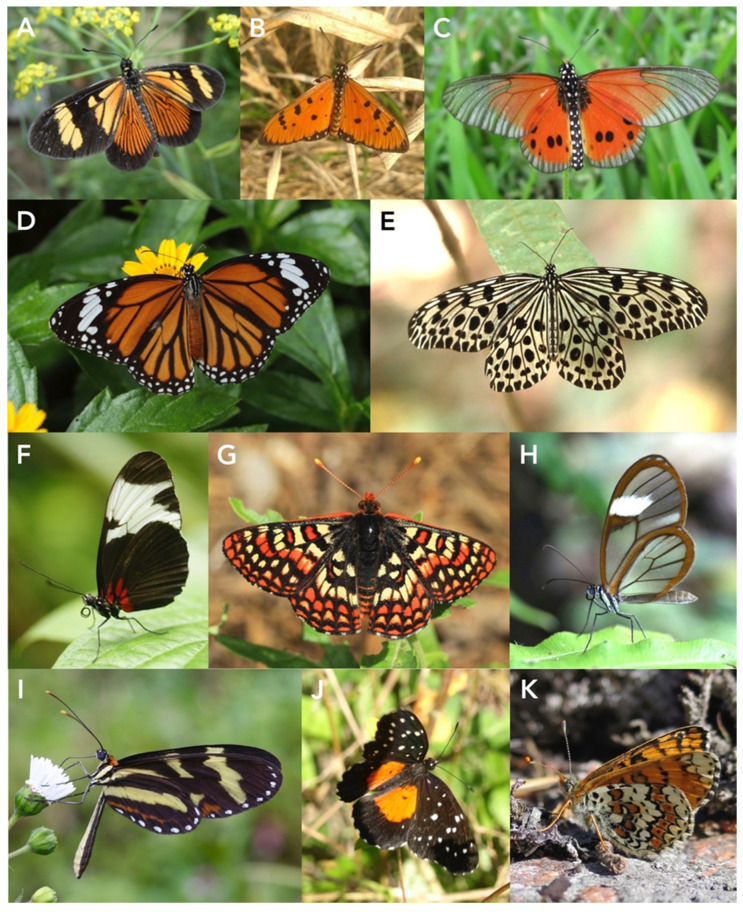
Exemplar adults of toxic Nymphalidae. (**A**) *Actinote pellenea* (Acraeinae) from South America (reproduced under terms of Creative Commons Attribution License 3.0, credit Dianakc); (**B**) tawny coster, *Acraea terpsicore* (Acraeinae), from Thailand (reproduced under terms of Creative Commons Attribution-ShareAlike 4.0, credit Soumyapatra13); (**C**) *Acraea igola* (Acraeinae) from Mozambique (reproduced under terms of Creative Commons Attribution Share Alike 2.0, credit Ton Rulkens); (**D**) common tiger, *Danaus genutia* (Danainae), from Thailand; (**E**) tree nymph, *Idea lynceus* (Danainae), from Thailand; (**F**) *Heliconius sapho* (Heliconiinae) from Costa Rica (reproduced under terms of Creative Commons Attribution-ShareAlike 4.0, credit Hans Hillewaert); (**G**) Edith’s checkerspot, *Euphydryas editha* (Heliconiinae), from California (reproduced under terms of Creative Commons Attribution-ShareAlike 2.0, credit Judy Gallagher); (**H**) *Episcada polita* (Ithomiinae) from Ecuador (reproduced under terms of Creative Commons Attribution-ShareAlike 2.0, credit Dr. Alexey Yakovlev); (**I**) *Mechanitis menapis mantineus* (Ithomiinae) from Ecuador (reproduced under terms of Creative Commons Attribution-Share Alike 2.0, credit Dr. Alexey Yakovlev); (**J**) crimson patch, *Chlosyne janais* (Ithomiinae), from Mexico (reproduced under terms of Creative Commons Attribution-ShareAlike 3.0, credit Beatriz Moisset); (**K**) *Melitaea cinxia* (Melitaeini) from Finland (photograph by and © Saskya van Nouhuys, reproduced with permission).

**Figure 4 toxins-15-00424-f004:**
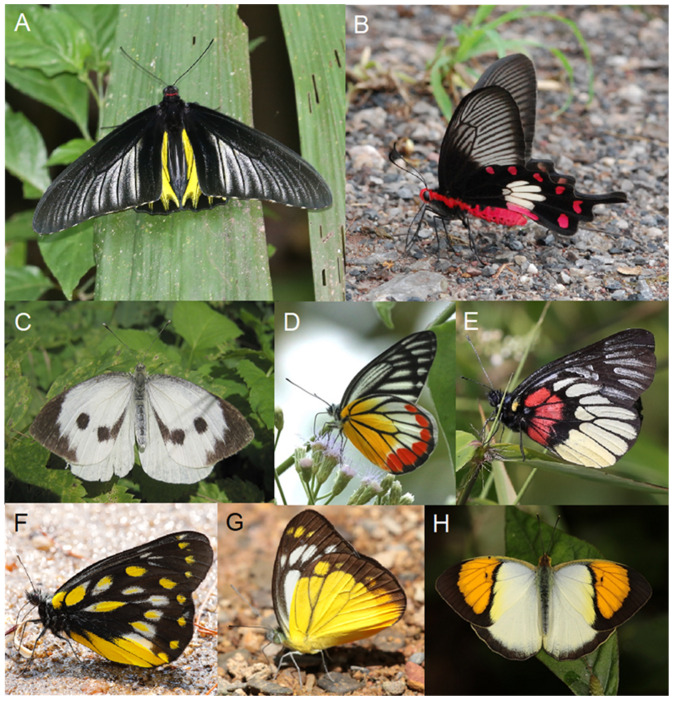
Exemplar adults of toxic species of papilionid and pierid butterflies. (**A**) *Troides* sp., possibly *T. aeacus*, (Papilionidae: Troidini) from Thailand; (**B**) *Pachliopta aristolochiae* (Papilionidae: Troidini) from Thailand; (**C**) large white, *Pieris brassicae* (Pieridae: Pierini) (photograph by S. Sepp, reproduced under the terms of GNU Free Documentation License); (**D**) painted jezebel, *Delias hyparcte*, male (Pieridae: Pierini); (**E**) red-breast jezebel, *Delias acalis* (Pieridae: Pierini); (**F**) dark jezebel, *Delias berinda* (Pieridae: Pierini), from Thailand; (**G**) orange gull, *Cepora iudith* (Pieridae: Pierini), from Thailand; (**H**), yellow orange tip, *Ixias pyrene* (Pieridae: Teracolini), from Thailand.

**Figure 5 toxins-15-00424-f005:**
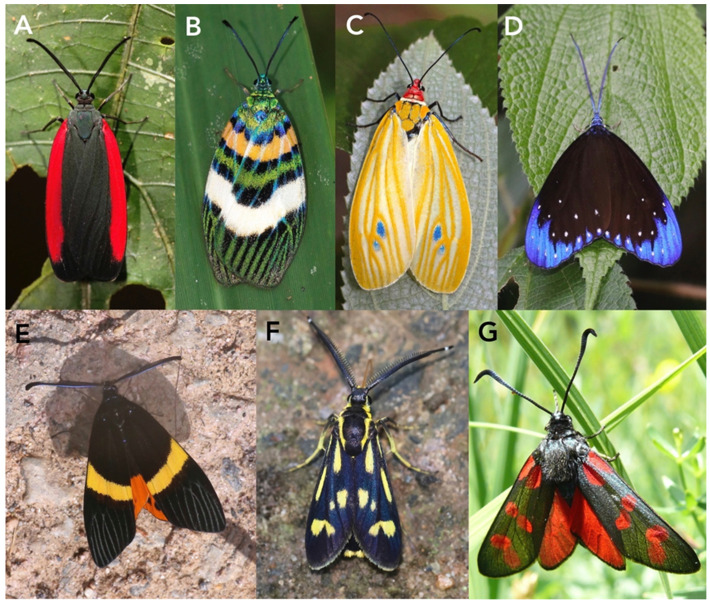
Exemplar adults of toxic species of Zygaenidae. (**A**) *Retina rubrivittas* (Chalcosiinae) from Thailand; (**B**) *Chalcophaedra zuleika* (Chalcosiinae) from Thailand; (**C**) *Soritia leptalina* from Thailand (Chalcosiinae); (**D**) *Cyclosia midama* (Chalcosiinae) from Thailand; (**E**) *Pidorus yayoiae* (Chalcosiinae) from Thailand; (**F**) *Artona zebra* (Procridinae) from Thailand; (**A**–**F** photographs © Antonio Giudici, reproduced with permission); (**G**) *Zygaena filipendulae* (Zygaeninae) from Germany (reproduced under terms of Creative Commons Attribution-ShareAlike 2.5, credit Harald Süpfle).

**Figure 6 toxins-15-00424-f006:**
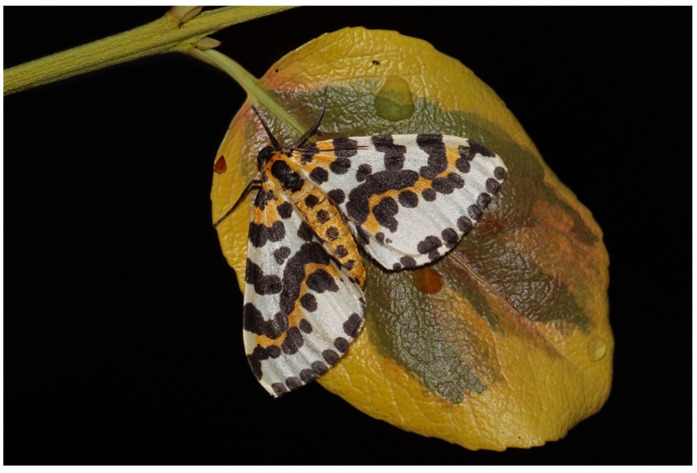
Adult magpie moth, *Abraxas grossulariata* (Geometridae) (reproduced under terms of Creative Commons Attribution License 4.0, credit Sharp Photography, sharpphotography).

**Figure 7 toxins-15-00424-f007:**
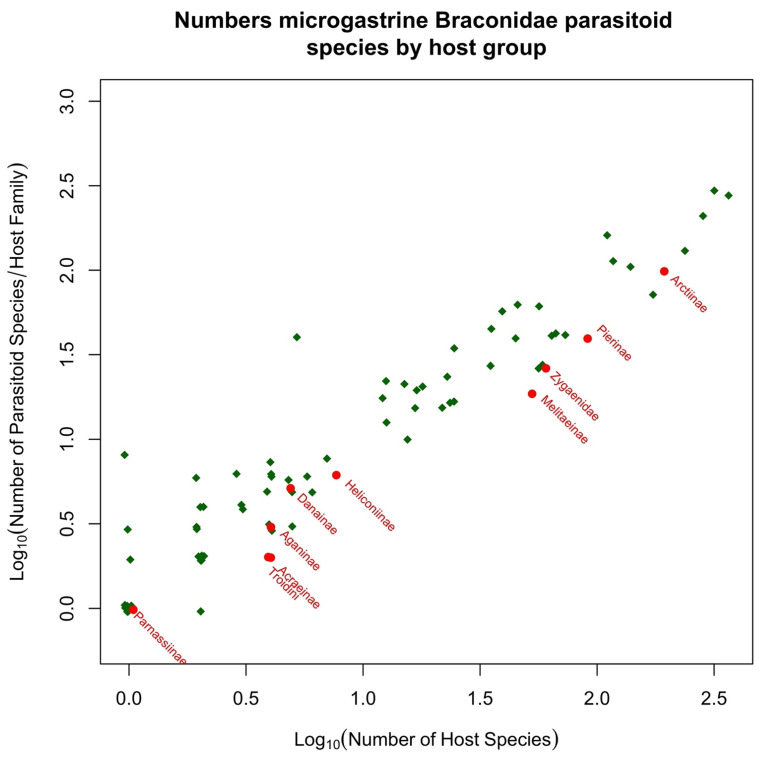
Relationship between the number of recorded host species of microgastrine braconids within a given family or other distinguished group of Lepidoptera (x-axis) and the number of unique parasitoid host species combinations in each group (y-axis). Numbers are presented on logarithmic scales, and the points representing the nine largely chemically protected host groups are indicated in red. All points have been jittered by up to 2% to distinguish coincident points, especially towards the origin of the graph.

**Figure 8 toxins-15-00424-f008:**
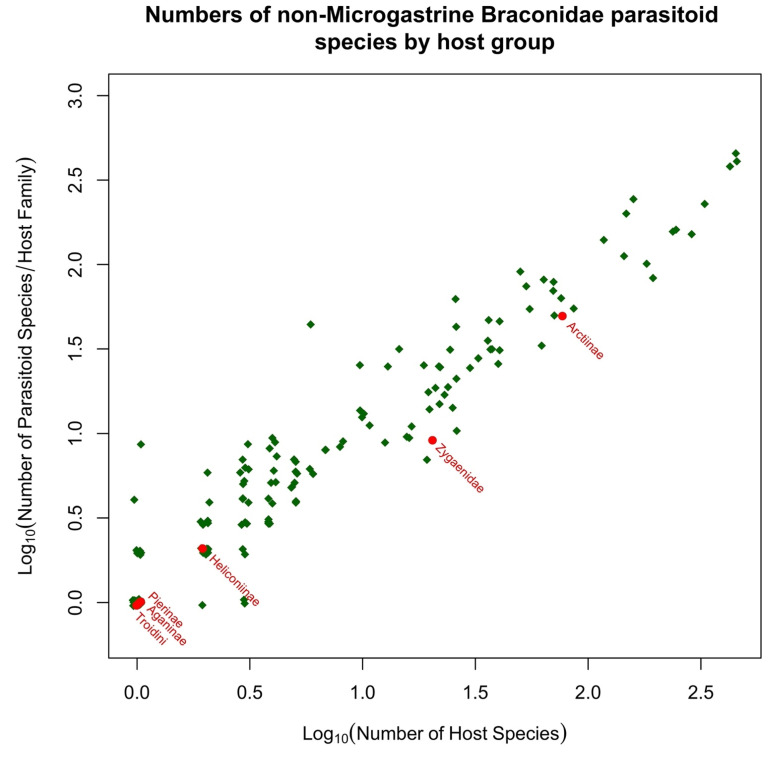
Relationship between the number of recorded host species of non-microgastrine braconids within a given family or other distinguished group of Lepidoptera (x-axis) and the number of unique parasitoid host species combinations in each group (y-axis). See legend of Figure 7 for further details.

**Figure 9 toxins-15-00424-f009:**
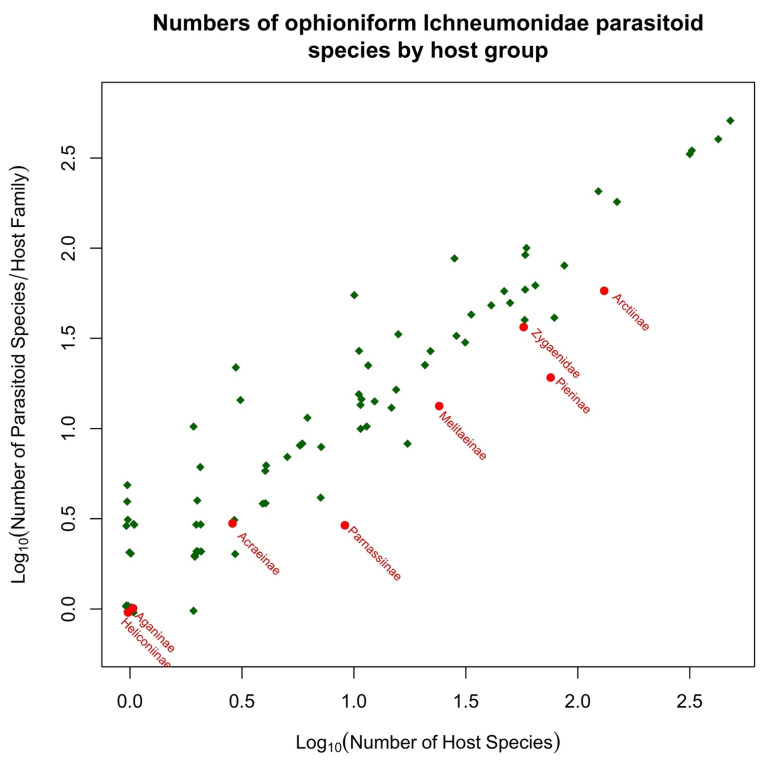
Relationship between the number of recorded host species of ‘ophioniformes’ ichneumonids within a given family or other distinguished group of Lepidoptera (x-axis) and the number of unique parasitoid host species combinations in each group (y-axis). See legend of Figure 7 for further details.

**Figure 10 toxins-15-00424-f010:**
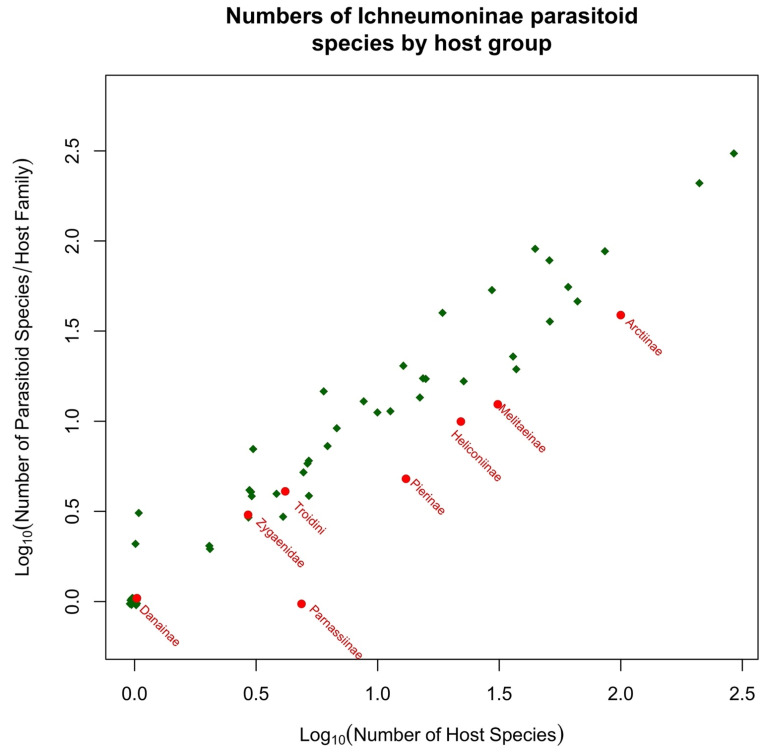
Relationship between the number of recorded host species of Ichneumoninae within a given family or other distinguished group of Lepidoptera (x-axis) and the number of unique parasitoid host species combinations in each group (y-axis). See legend of Figure 7 for further details.

**Figure 11 toxins-15-00424-f011:**
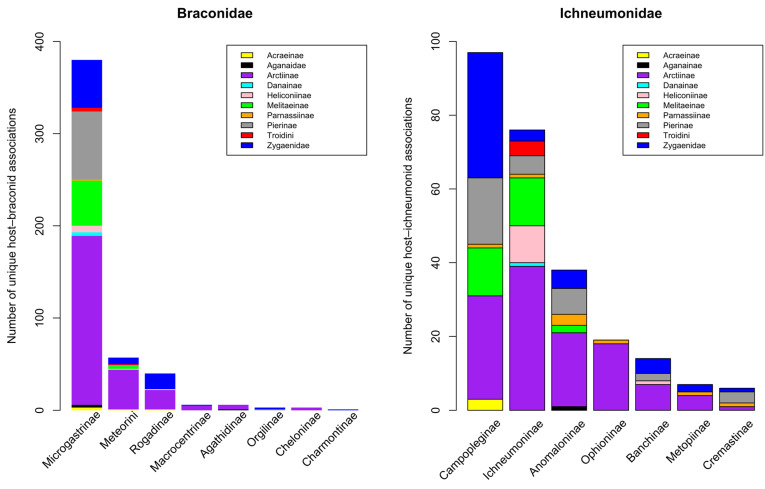
Summary of the number of unique host–parasitoid associations separately for selected Braconidae and Ichneumonidae that parasitize predominantly folivorous Lepidoptera.

**Table 3 toxins-15-00424-t003:** Number of unique ichneumonid wasp host associations for taxa specialized on Lepidopteran hosts (Source: data from Yu et al. [149]).

Subfamily (Total Number of Published Host Records)	Number of Associations with Palatable Host Groups	Number of Associations with Unpalatable Host Groups	Proportion of Associations Involving Unpalatable Hosts
Anomaloninae (1017)	921	59	0.064
Banchinae (1303)	1231	17	0.0138
Campopleginae (4192)	3485	156	0.0448
Cremastinae (734)	662	5	0.0076
Ichneumoninae (2896)	2639	159	0.0602
Metopiinae (676)	621	9	0.0145
Ophioninae (633)	607	23	0.0145

## Data Availability

The data presented in this study are available from the corresponding author upon reasonable request.

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
