# Peer review of "Dietary Challenges for Parasitoid Wasps (Hymenoptera: Ichneumonoidea); Coping with Toxic Hosts, or Not?"

_toxins, 2023, doi:10.3390/toxins15070424_

Round 1

Reviewer 1 Report

The topic well described in the manuscript, and the work complements previous knowledge and opens up the possibility for further research. The study is very interesting. The article is very well written and organized. The authors did an informative comparative study, some points should be revised before final publications.

-    Table 2, 3: the title of 2nd and 3th column are the same, revise.
-    Some minor typo error should be adjusted.

Author Response

Dear the reviewer,

Thank you very much for the comments and your valuable time. We are pleased that you like the MS. We did the corrections according to your comments and suggestion. 

  • Table 2, 3: the title of 2nd and 3th column are the same, revise. [Corrected]
  • Some minor typo error should be adjusted. [Done].

Thank you very much.

Best regards,

Authors. 

Reviewer 2 Report

In this review, the dietary challenges faced by parasitoids (in particular ichneumonoids) when encountering toxic hosts are dealt with. In the overall, I appreciated the paper and found it very interesting. However, I find it exceedingly long. In some parts, the paper should be more focused on the effects of plant/host toxin on the parasitoid complex of the target insect hosts. Namely, in the “Toxin” section, paragraphs dealing with host/plant toxins, with no reference to parasitoids, might be condensed (see specific comments). The “host” section also needs to be condensed and focused on the topics mainly related with the host/parasitoid relationship. I also suggest condensing the “parasitoid” section, by reducing the information about taxonomy, which I find redundant in this kind of review, and not that suitable for this journal.

Yhe Matrials and Methods section should be anticipated before the results

The pictures of butterflies/moths are nice, but they are not really necessary. They may be reduced (especially Figure 3), if not deleted.

The overall language is good, however, some sentences need to be verified, because they lack verb or are not clear. I have indicated some of them in the comments below,, but the overall text heed to be checked, also for some mistakes in spelling.

Here below there are some comments, some of which are considerations that arose from reading the text, which the authors might consider (for this or future work)

Line 18: “can be toxic”

Lines 63-67: the population size of the wasps may also depend on the degree of polyphagy of the wasps

Lines 73-76: the capacity of parasitoids to manipulate the host physiology and behaviour is not related to the toxic compounds present in the host

Lines 136-140: does any of the hosts of these generalist Enicospilus species produce toxins?

Lines 140-142: may the abiotic factors have an influence on these “generalist” parasitoid distribution?

Lines 201-203: interesting hypothesis. Compsilura also displays a high capacity to avoid host encapsulation  (which results in its high polyphagy) and I wonder if this may be related to the parasitoid’s ability to escape host’s toxins

Lines 252-254: this sentence is not clear to me, please re-formulate

Line 275: I assume it is “synthesise”

Lines 280-292: I suggest considering to condense

Line 324: May you give some example of parasitoid species attacking P. xylostella, especially Ichneumonids?

Line 353: “investigated by Boros et al. [85]”

Lines 353-359: do these large quantities of IG antirrinoside have an influence on the parasitoid complex of M. paradoxa?

Lines 383-386: In my opinion these two sentence may be condensed

Line 390: “an” acceptable…

Line 397: is there any effect on parasitoids?

Lines 412-13: please re-formulate the sentence

Line 427: “The host”

Line 505: it would be fair if any parasitoid species is mentioned

Lines 509-512: even if the parasitoid larva is killed, it may have the time to negatively affect the (though unsuitable) host, i.e., its  survival and development. The death of the parasitoid, either for toxins or encapsulation, has to be very precocious not to affect the host.

Lines 522-525: please re-formulate the sentence and specify what Taxapad is (not everybosy is familiar with this database)

Lines 526-528: it seems to me that the verb is missing

Line 534: which tachinid?

Line 553: “caterpillar”

Lines 557-558:  “but the fact that most have not, simply means that nobody has reared them or if they have, bothered to report any parasitoids”: this is a speculation, delete.

Line 564: delete commas before and after the species name

Lines 568-570: this sentence is not clear to me, please re-formulate

Lines 576-583: please condense

Lines 586-591: this sentence is too long and not fully clear. Split in two

Line 608: replace “is” with “are”

Line 611: “Holzinger et al. [158]

Lines 652-653: delete the sentence in brackets.

Line 672: “are some”

Line 712: “diplays”

Line 743: please complete the sentence.

Lines 751-752: “The lymantriid Ivela auripes [182]. All these moths have conspicuously coloured larvae and pupae which are formed in exposed positions”. This sentence needs to be re-formulated

Lines 765-779: I suggest condensing

Lines 787-791: please re-formulate the sentence

Line 817: Hyphantria cunea was previously mentioned. The genus name should be abbreviated

Line 874: “may confer”

Line 893: may be “in the informal Cotesia group”?

Line 896: “and have received”

Lines 905-6: please check the sentence

Lines 912-914 and 927: Please check the sentence

Lines 1058 and following: this interesting information on Taxapad should be given before. Taxapad is mentioned throughout the paper well before this section

Lines 1100-1103: please check the sentence

Lines 1116 and following: in Table 1, probably the third column heading is “Number of associations with palatable host groups

Lines 1241 and following: Polydnaviruses are produced by ichneumonoid wasps that are not necessarily parasitoids of toxic hosts. Their implication in the host's toxin production or sequestration mechanisms is speculative, but it may be hypothesized

Lines 1241-44: please check the sentence

Author Response

Dear the reviewer,

Thank you very much for your comments and suggestion. We appreciated your valuable time. 

We did the corrections and add a bit of our comments in [      ].

Please let's us know if there is anything else we should edit to make this MS better.

Thank you very much.

Best regards,

the Authors

In this review, the dietary challenges faced by parasitoids (in particular ichneumonoids) when encountering toxic hosts are dealt with. In the overall, I appreciated the paper and found it very interesting. [We are glad the referee generally likes it.] However, I find it exceedingly long. [We see the purpose of a review article not to be for those who are already experts in the field, but for those wishing to get into the field, such as starting PhD students or postdocs, and therefore we consider all the topics covered to be highly relevant, and believe that citations to papers provided in each area will be very helpful to such readers.] In some parts, the paper should be more focused on the effects of plant/host toxin on the parasitoid complex of the target insect hosts. Namely, in the “Toxin” section, paragraphs dealing with host/plant toxins, with no reference to parasitoids, might be condensed (see specific comments). [We have done some re-writing and cross-referencing.]

The “host” section also needs to be condensed and focused on the topics mainly related with the host/parasitoid relationship. I also suggest condensing the “parasitoid” section, by reducing the information about taxonomy, which I find redundant in this kind of review, and not that suitable for this journal. [We disagree because many of those scientists working on the toxins in these systems have a predominantly biochemical background, and we strongly believe that the biological and phylogenetic relationships of the parasitoids are important for understanding the evolution of the systems. The aim of a review article is to provide a broad overview and point to all relevant literature, not only/really for those already working in the field, but, for example, to postgrads and postdocs entering the field afresh.]

The Materials and Methods section should be anticipated before the results.  [The order of sections, follow the template of /Toxins/].

The pictures of butterflies/moths are nice, but they are not really necessary. They may be reduced (especially Figure 3), if not deleted. [We disagree and think they add to the idea that these toxic hosts are very warningly coloured – again readers coming from chemical/biochemical backgrounds may not be very familiar with the insects and their natural history such as Batesian and Mullerian mimicry. Further, Prof. Prevost, the editor of this special issue who invited us, is happy with it. Fig. 3 is actually one of the more interesting ones. To humans some of these butterflies might not look particularly aposematic, but it is clear that there are elements of their colour and pattern that birds do learn are associated with unpalatability.]

The overall language is good, however, some sentences need to be verified, because they lack verb or are not clear. [Done.] I have indicated some of them in the comments below, but the overall text heed to be checked, also for some mistakes in spelling. [We have rechecked and made some corrections.]

Here below there are some comments, some of which are considerations that arose from reading the text, which the authors might consider (for this or future work) [Our thanks for the referee's attention.]

Line 18: “can be toxic” [Done.]

Lines 63-67: the population size of the wasps may also depend on the degree of polyphagy of the wasps [Good point, we have added this.]

Lines 73-76: the capacity of parasitoids to manipulate the host physiology and behaviour is not related to the toxic compounds present in the host [We do not say that they do; this is an uninvestigated area. What we are trying to provide the reader with, is a broad context.]

Lines 136-140: does any of the hosts of these generalist Enicospilus species produce toxins? [We added a clause saying that, apart from toxins in their defensive spines, they do not sequester plant toxins.]

Lines 140-142: may the abiotic factors have an influence on these “generalist” parasitoid distribution? [This seems unlikely to us and we are not aware of any suggestions that this is the case as regards the insects. In the NHH, it has been postulated, based on considerable evidence, that plants in the tropics are on average more toxic and one explanation might be that higher light levewls and temperatures might make it relatively less expensive to synthesise toxins.]

Lines 201-203: interesting hypothesis. Compsilura also displays a high capacity to avoid host encapsulation (which results in its high polyphagy) and I wonder if this may be related to the parasitoid’s ability to escape host’s toxins. [This does not appear to be known so we cannot comment.]

Lines 252-254: this sentence is not clear to me, please re-formulate [We have explained it more fully.]

Line 275: I assume it is “synthesise” [Corrected.]

Lines 280-292: I suggest considering to condense [We have not changed this as we consider it important basic information on the toxicity of fouranocoumarins.]

Line 324: May you give some example of parasitoid species attacking P. xylostella, especially Ichneumonids? [Added.]

Line 353: “investigated by Boros et al. [85]” [Added.]

Lines 353-359: do these large quantities of IG antirrinoside have an influence on the parasitoid complex of M. paradoxa? [We added sentence that it has no known parasitoids.]

Lines 383-386: In my opinion these two sentence may be condensed [We have merged into one sentence and shortened it.]

Line 390 [394]: “an” acceptable… [Corrected.]

Line 397: is there any effect on parasitoids? [This point is dealt with elsewhere. This sentence was just about the use of PAs as sex pheromone precursors not for protection.]

Lines 412-13: please re-formulate the sentence [We have re-written and split in to two sentences it to make it clearer.]

Line 427: “The host” [Done.]

Line 505 [509]: it would be fair if any parasitoid species is mentioned [We have added the parasitoid name and explained more clearly what the cited paper was referring to.]

Lines 509-512: even if the parasitoid larva is killed, it may have the time to negatively affect the (though unsuitable) host, i.e., its  survival and development. The death of the parasitoid, either for toxins or encapsulation, has to be very precocious not to affect the host. [Yes, this is true, we don't say otherwise]

Lines 522-525: please re-formulate the sentence and specify what Taxapad is (not everybosy is familiar with this database) [Added.]

Lines 526-528: it seems to me that the verb is missing [We have re-written the sentence to correct its grammar.]

Line 534: which tachinid? [We have added the details of the tachinid flies.]

Line 553: “caterpillar” [Added.]

Lines 557-558:  “but the fact that most have not, simply means that nobody has reared them or if they have, bothered to report any parasitoids”: this is a speculation, delete. [We have reworded this but think it important to point out the general dearth of such data. Indeed there are not even any foodplant records for the majority of tropical butterflies, and even fewer for moths.]

Line 564: delete commas before and after the species name [Done.]

Lines 568-570: this sentence is not clear to me, please re-formulate [We have re-written the last part of this paragraph to make it clearer.]

Lines 576-583: please condense [Shortened a bit, some redundancy removed.]

Lines 586-591: this sentence is too long and not fully clear. Split in two [Split as requested and clarified.]

Line 608: replace “is” with “are” [Done.]

Line 611: “Holzinger et al. [158] [Added.]

Lines 652-653: delete the sentence in brackets. [Done.]

Line 672: “are some” [Corrected.]

Line 712: “diplays” [Corrected.]

Line 743: please complete the sentence. [Done.]

Lines 751-752: “The lymantriid Ivela auripes [182]. All these moths have conspicuously coloured larvae and pupae which are formed in exposed positions”. This sentence needs to be re-formulated [Rewritten.]

Lines 765-779: I suggest condensing [Shortened and partly re-written.]

Lines 787-791: please re-formulate the sentence [Completely re-written to claify and add references.]

Line 817: Hyphantria cunea was previously mentioned. The genus name should be abbreviated [Done.]

Line 874: “may confer” [Corrected.]

Line 893: may be “in the informal Cotesia group”? [Done.]

Line 896: “and have received” [Corrected.]

Lines 905-6: please check the sentence [We have re-written the sentence to make it more explicit what we mean.]

Lines 912-914 [Corrected.] and 927: Please check the sentence [Modified.]

Lines 1058 and following: this interesting information on Taxapad should be given before. Taxapad is mentioned throughout the paper well before this section [Yes, but this is difficult because Journal style is that the materials and methods should be placed at the end of the MS. we followed journal style.]

Lines 1100-1103: please check the sentence [Superfluous word removed, and grammar corrected.]

Lines 1116 and following: in Table 1, probably the third column heading is “Number of associations with palatable host groups [Done as per other referee's comment.]

Lines 1241 and following: Polydnaviruses are produced by ichneumonoid wasps that are not necessarily parasitoids of toxic hosts. Their implication in the host's toxin production or sequestration mechanisms is speculative, but it may be hypothesized [We only hypothesised that they might be, and pointed to the complete lack of information when it comes to toxic hosts.]

Lines 1241-44: please check the sentence [The sentence seems to be fine to us. Stet.]

Reviewer 3 Report

Major revision

1.      The authors gave many representative pictures about the toxic species. However, in part 2, the authors stated many kinds of toxics derived from plants, we hope we can see the pictures of toxics such as molecular structure but not host species pictures.  

2. This manuscript was submitted as a review article but it included material and methods at the end of manuscript. It looks like system review article but it did not described as the protocol of systematic reference review article. The design makes me confused.  

3.      From figure 2 to 6, it is not necessary to take four figures to state the different host.

4.  The font resolution of figure 7 to 10 was not clear.

I have no much suggestion on language.

Author Response

Dear the reviewer,

Thank you very much for you valuable comments, suggestion and time. We are appreciated it. 

We did the corrections according to your comments, we are explain each of the comment in [        ].

If you have further comments, please do let's us know ?

Thank you very much.

Best regards,

the Authors

  1. The authors gave many representative pictures about the toxic species. However, in part 2, the authors stated many kinds of toxics derived from plants, we hope we can see the pictures of toxics such as molecular structure but not host species pictures. [We are rather unclear what this referee is asking for. In this point they seem to be asking that we show the chemical structures of [representatives of] all or most of the various groups of toxins that we mention. We can of course add these but it would make the MS quite a lot longer, even if several a grouped together in single plates. At least two full page figures would be required. Please let us know whether you would like us to add molecular structure diagrams which we can easily do, and if so, ca the Journal editors agree that this is desirable?]
  2. This manuscript was submitted as a review article but it included material and methods at the end of manuscript. It looks like system review article but it did not described as the protocol of systematic reference review article. The design makes me confused. [We stated clearly in the letter with our original submission, that although this is largely a review article which had been 'commissioned', that to do a proper job it actually required original analyses of published host relationships that had never previously been considered in this respect. Therefore, our MS combines both a literature review and a metanalysis of published data.]
  3. From figure 2 to 6, it is not necessary to take four figures to state the different host. [We think it is important to show a range of representatives of ALL of the major groups of toxic hosts. Since the MS if published would be electronic, and since the images might well be expected to attract interest from readers, we see no harm in showing them for comparative purposes. Since only individually paid-for reprints are actually printed in hard copy there are no significant production cost implications, and we believe that nice illustrations will increase the chances of the article being cited which is good for us and good for the journal.]
  4. The font resolution of figure 7 to 10 was not clear. [Yes, this appears to be because of the embedding of pdf within pdf. We will replace with .tiff for final version.]

Round 2

Reviewer 2 Report

The Authors well replied to all my comments. I still think that the review (that I found very interesting in the overall) is long, but, if it is also considered as a means to get familiar with the topic for PhD students/early researchers (as stated by the authors) it is appropriate. Since the Editor appreciated all photos and agreed to insert all of them in the paper, I agree as well. I still think that the Materials and Methods should not be placed at the end of the paper, but, apparently, this was not the authors' choice.

Reviewer 3 Report

it can be accepted as the current version.